# TGF-β-dependent reprogramming of amino acid metabolism induces epithelial–mesenchymal transition in non-small cell lung cancers

Fumie Nakasuka [1,2,3,4,16], Sho Tabata [1,2,5,6,16 ✉], Takeharu Sakamoto [7,8], Akiyoshi Hirayama [1,2], Hiromichi Ebi [9], Tadaaki Yamada[10], Ko Umetsu[1], Maki Ohishi[1], Ayano Ueno[1], Hisatsugu Goto[11], Masahiro Sugimoto[1,2,12], Yasuhiko Nishioka[11], Yasuhiro Yamada[3,4], Masaru Tomita[1,2], Atsuo T. Sasaki[1,5,13,14], Seiji Yano[15] & Tomoyoshi Soga[1,2]

Epithelial–mesenchymal transition (EMT)—a fundamental process in embryogenesis and wound healing—promotes tumor metastasis and resistance to chemotherapy. While studies have identified signaling components and transcriptional factors responsible in the TGF-β-dependent EMT, whether and how intracellular metabolism is integrated with EMT remains to be fully elucidated. Here, we showed that TGF-β induces reprogramming of intracellular amino acid metabolism, which is necessary to promote EMT in non-small cell lung cancer cells. Combined metabolome and transcriptome analysis identified prolyl 4-hydroxylase α3 (P4HA3), an enzyme implicated in cancer metabolism, to be upregulated during TGF-β stimulation. Further, knockdown of P4HA3 diminished TGF-β-dependent changes in amino acids, EMT, and tumor metastasis. Conversely, manipulation of extracellular amino acids induced EMT-like responses without TGF-β stimulation. These results suggest a previously unappreciated requirement for the reprogramming of amino acid metabolism via P4HA3 for TGF-β-dependent EMT and implicate a P4HA3 inhibitor as a potential therapeutic agent for cancer.

[1] Institute for Advanced Biosciences, Keio University, Tsuruoka, Yamagata, Japan. [2] Systems Biology Program, Graduate School of Media and Governance, Keio University, Fujisawa, Kanagawa, Japan. [3] Department of Computational Biology and Medical Sciences, Graduate School of Frontier Sciences, The University of Tokyo, Chiba, Japan. [4] Division of Stem Cell Pathology, Center for Experimental Medicine and Systems Biology, Institute of Medical Science, The University of Tokyo, Tokyo, Japan. [5] Division of Hematology and Oncology, Department of Internal Medicine, University of Cincinnati, Cincinnati, OH, USA. [6] Institute for Protein Research, Osaka University, Suita, Osaka, Japan. [7] Division of Cellular and Molecular Biology, The Institute of Medical Science, The University of Tokyo, Minato-ku, Tokyo, Japan. [8] Department of System Biology, Institute of Medical, Pharmaceutical and Health Sciences, Kanazawa University, Kanazawa, Ishikawa, Japan. [9] Division of Molecular Therapeutics, Aichi Cancer Center Research Institute, Nagoya, Aichi, Japan. [10] Department of Pulmonary Medicine, Kyoto Prefectural University of Medicine, Kyoto, Japan. [11] Department of Respiratory Medicine and Rheumatology, Graduate School of Biomedical Sciences, Tokushima University, Tokushima, Tokushima, Japan. [12] Health Promotion and Preemptive Medicine, Research and Development Center for Minimally Invasive Therapies, Tokyo Medical University, Shinjuku, Tokyo, Japan. [13] Department of Cancer Biology, University of Cincinnati College of Medicine, Cincinnati, OH, USA. [14] Department of Neurosurgery, Brain Tumor Center at UC Gardner Neuroscience Institute, Cincinnati, OH, USA. [15] Department of Medical Oncology, Kanazawa University Cancer Research Institute, Kanazawa University, Kanazawa, Ishikawa, Japan. [16] These authors contributed equally: Fumie Nakasuka, Sho Tabata. ✉email: tabatasho@gmail.com

Epithelial–mesenchymal transition (EMT) is a reversible differentiation process during which epithelial cells acquire mesenchymal properties. It is characterized by the loss of cell–cell adhesion and conversion from a cobblestone-like to more spindle-shaped morphology, enhancing cell motility and invasion. EMT is an essential, fundamental process during the developmental stages of metazoan, and is also activated during the development of organ fibrosis and wound healing[1–3]. Activation of EMT in solid tumor cells increases the rate of local, as well as distal, metastases. In addition, EMT has been shown to enhance acquisition of a cancer stem-like phenotype, confer resistance to chemotherapeutic drugs, promote cell survival under stressed conditions, such as reactive oxygen species elevation and nutrient deprivation, and induce immunosuppressive tumor microenvironments[1–3]. In particular, EMT is clinically recognized as associated with disease progression and poor prognosis in non-small-cell lung cancer (NSCLC)[4,5]. Despite advances in available therapies, the prognosis of NSCLC remains poor with a 5-year overall survival rate of <20%, due to its highly invasive and metastatic features[6].

Transforming growth factor-β (TGF-β) is one of the best-known potent inducers of EMT. In contrast to its name, TGF-β is also known to suppress cancer cell proliferation in a cell context-dependent manner[7]. TGF-β ligand binds to its cognate receptors expressed on the cell surface, in an autocrine or paracrine fashion, and activate a family of SMAD transcriptional factors, leading to upregulation of SNAI1 (SNAIL1), SNAI2 (SNAIL2/SLUG), TWIST, and ZEB1—master transcriptional factors that induce EMT. Extensive studies have revealed how these EMT-transcriptional factors alter the transcriptional landscape to promote EMT during the development and tumor progression[8–10].

Several tumors increase the rate of protein, lipid, and nucleotide synthesis for their malignant growth. Moreover, metabolic reprogramming is significantly associated with the early onset of tumorigenesis in the vast majority of cancers, and enhances the synthesis of macromolecules, including proteins, lipids, and nucleotides, to meet the increased metabolic demand of highly proliferating cells[11–15]. Of particular interest, compared to those analyzed in tumors during the proliferative stage, EMT represents a unique situation in which cell proliferation status is suppressed and factors associated with tumor malignancy, such as cellular motility and stemness, are elevated. Accumulating evidences have shown the linkage of EMT to cellular metabolism, and its functional importance in cancer metastasis[16–20]. Specifically, recent studies have reported that the loss of fumarate hydratase and subsequent accumulation of fumarate in renal cancer cells[21], as well as the increase in asparagine bioavailability via asparagine synthetase in metastatic breast cancer[22], and the increased expression of dihydropyrimidine dehydrogenase leading to dihydropyrimidine production in breast cancer cells[23] contributes to the acquisition of mesenchymal characteristics in vitro and in vivo. Cumulatively, these findings suggest that the bioavailability of certain metabolites is critical for inducing EMT. However, there are few mechanistic studies of EMT using comprehensive metabolomics combined with transcriptome analysis. It is critical to investigate the potential role of the other metabolic pathways altered after the EMT of cancer cells. Accordingly, in the current study, we identified the reprogramming of amino acid metabolism during TGF-β-induced EMT through a comprehensive metabolic analysis and demonstrated its significance to EMT phenotypes.

## Results

**TGF-β induces metabolic reprogramming in NSCLC cells.** To investigate the effect of TGF-β-induced EMT on metabolism in NSCLC cells, three well-studied NSCLC cell lines (A549, HCC827, and H358) were treated with TGF-β for 3 days as well as several weeks, and polar metabolites were quantified by capillary electrophoresis time-of-flight mass spectrometry (CE-TOFMS)[24] (Supplementary Data 1–3). TGF-β-induced EMT of NSCLC cells was verified by the morphological changes and expression of EMT marker genes E-cadherin (CDH1), N-cadherin (CDH2), fibronectin 1 (FN1), matrix metallopeptidase (MMP)-9, and MMP-2 (Supplementary Figs. 1a–f). We performed principal component analysis (PCA) and metabolic pathway enrichment analysis (MPEA) for the metabolites altered upon TGF-β treatment (Supplementary Figs. 1g–i and Supplementary Data 4–6). Based on MPEA, 21 pathways were changed by TGF-β in the cell lines (Fig. 1a), among which, alteration of amino acid metabolism was commonly detected (Fig. 1a, right).

To examine the link between amino acid metabolism and EMT in detail, we conducted a time-course metabolome analysis on A549 cells (Supplementary Fig. 1j and Supplementary Data 7). Intriguingly, TGF-β treatment for 72 h increased the levels of Asp, Glu, and Lys whereas levels of Ala, Asn, citrulline, Gln, Gly, His, hydroxyproline, Ile, Leu, Phe, Pro, Thr, and Tyr were decreased (Fig. 1b and Supplementary Fig. 1k). Figure 1c shows the amino acid metabolism mapped to the altered amino acids.

TGF-β-induced EMT is a reversible process. After TGF-β treatment in A549 cells, TGF-β withdrawal resulted in mesenchymal-to-epithelial transition (MET), which is the opposite to EMT (Fig. 2a, upper)[20] and is required for metastatic colonization in distant tissues[25]. The mRNA level of CDH1 (epithelial marker) was decreased, and that of CDH2 and FN1 (mesenchymal markers) increased upon TGF-β stimulation (Fig. 2a, lower). In contrast, these responses were quickly reversed after TGF-β withdrawal. Then, we examined whether the levels of amino acids were reversibly regulated by TGF-β withdrawal. The observed amino acid changes in EMT cells were reversed after MET, except for Asp (Fig. 2b and Supplementary Fig. 2). Notably, the decrease in amino acids by TGF-β was strongly correlated with mesenchymal status.

**Induction of EMT phenotypes by amino acid depletion.** Next, we examined whether the TGF-β-dependent changes in amino acids are the result of a passive phenomenon or are actively involved in the EMT process. Specifically, to investigate the effect of TGF-β-induced amino acid decline on EMT, we first prepared media depleted of Ala, Asn, Gly, His, hydroxyproline, Ile, Leu, Met, Phe, Pro, Thr, Trp, Tyr, and Val, thus mimicking the effect elicited by TGF-β (Figs. 1c, 2b), and examined the effect that the media on the expression of EMT markers (Fig. 3a, b, and Supplementary Fig. 3a). Treatment with media depleted of amino acids induced EMT-like responses, including down-regulation of CDH1 mRNA and upregulation of CDH2 and ZEB1 mRNA in A549 cells (Fig. 3a). However, although CDH1 protein abundance was reduced in a similar manner to CDH1 mRNA following treatment with amino acid depleted media, that of CDH2 and ZEB1 exhibited opposite trends to their corresponding mRNA expression, pointing to the posttranscriptional or posttranslational regulation[26–29] (Fig. 3b and Supplementary Fig. 3a).

Next, to further assess the role of amino acid depletion on EMT, we prepared media depleted of each of the 20 amino acids. A549 and SW1573 cells were subsequently cultured in each of these media for 72 h. Results show that depletion of Arg, Cys, Gln, or Met remarkably reduced cell growth in A549 cells (Supplementary Fig. 3b, c), whereas only Cys depletion impacted the growth of SW1573 cells (Supplementary Fig. 3b).

Additionally, depletion of a single amino acid (Phe, Thr, Trp, Lys, Val, Met, Leu, Ile, Gln, His, Arg, or Tyr) from the culture medium induced EMT-like patterns, without increasing CDH2

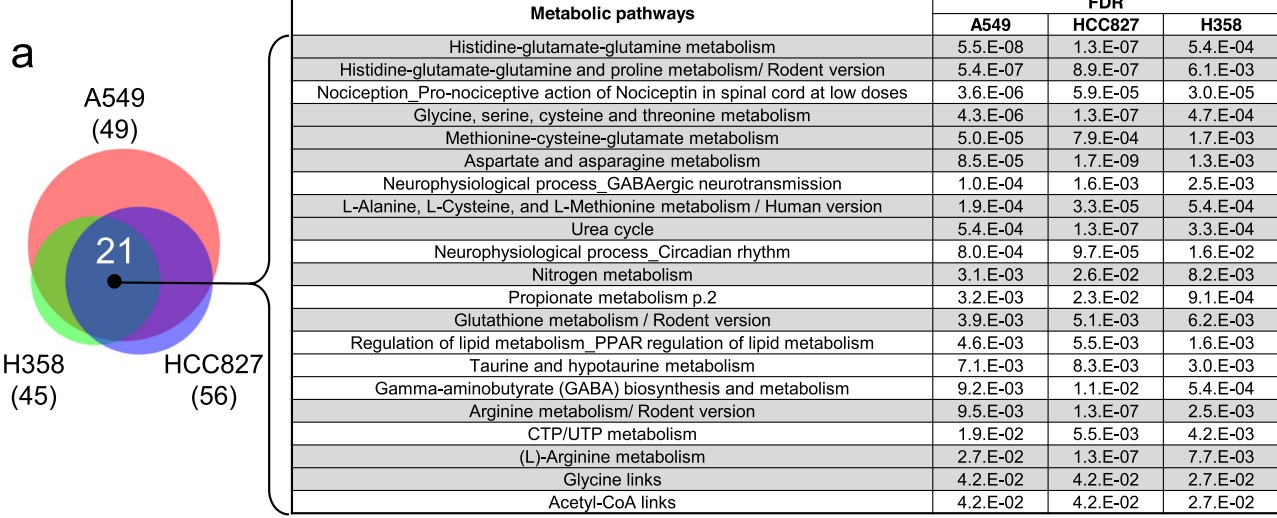

| Metabolic pathways | FDR | | |
|---|---|---|---|
| | A549 | HCC827 | H358 |
| Histidine-glutamate-glutamine metabolism | 5.5.E-08 | 1.3.E-07 | 5.4.E-04 |
| Histidine-glutamate-glutamine and proline metabolism/ Rodent version | 5.4.E-07 | 8.9.E-07 | 6.1.E-03 |
| Nociception_Pro-nociceptive action of Nociceptin in spinal cord at low doses | 3.6.E-06 | 5.9.E-05 | 3.0.E-05 |
| Glycine, serine, cysteine and threonine metabolism | 4.3.E-06 | 1.3.E-07 | 4.7.E-04 |
| Methionine-cysteine-glutamate metabolism | 5.0.E-05 | 7.9.E-04 | 1.7.E-03 |
| Aspartate and asparagine metabolism | 8.5.E-05 | 1.7.E-09 | 1.3.E-03 |
| Neurophysiological process_GABAergic neurotransmission | 1.0.E-04 | 1.6.E-03 | 2.5.E-03 |
| L-Alanine, L-Cysteine, and L-Methionine metabolism / Human version | 1.9.E-04 | 3.3.E-05 | 5.4.E-04 |
| Urea cycle | 5.4.E-04 | 1.3.E-07 | 3.3.E-04 |
| Neurophysiological process_Circadian rhythm | 8.0.E-04 | 9.7.E-05 | 1.6.E-02 |
| Nitrogen metabolism | 3.1.E-03 | 2.6.E-02 | 8.2.E-03 |
| Propionate metabolism p.2 | 3.2.E-03 | 2.3.E-02 | 9.1.E-04 |
| Glutathione metabolism / Rodent version | 3.9.E-03 | 5.1.E-03 | 6.2.E-03 |
| Regulation of lipid metabolism_PPAR regulation of lipid metabolism | 4.6.E-03 | 5.5.E-03 | 1.6.E-02 |
| Taurine and hypotaurine metabolism | 7.1.E-03 | 8.3.E-03 | 3.0.E-03 |
| Gamma-aminobutyrate (GABA) biosynthesis and metabolism | 9.2.E-03 | 1.1.E-02 | 5.4.E-04 |
| Arginine metabolism/ Rodent version | 9.5.E-03 | 1.3.E-07 | 2.5.E-03 |
| CTP/UTP metabolism | 1.9.E-02 | 5.5.E-03 | 4.2.E-03 |
| (L)-Arginine metabolism | 2.7.E-02 | 1.3.E-07 | 7.7.E-03 |
| Glycine links | 4.2.E-02 | 4.2.E-02 | 2.7.E-02 |
| Acetyl-CoA links | 4.2.E-02 | 4.2.E-02 | 2.7.E-02 |

3-PG: 3-Phosphoglyceric acid
α-KG: α-Ketoglutaric acid
P5C: 1-Pyrroline-5-carboxylate
HyPro: Hydroxyproline

**Fig. 1 Altered amino acid metabolism during TGF-β-induced epithelial–mesenchymal transition (EMT). a** Venn diagram representing metabolic pathways that were significantly altered in TGF-β-stimulated A549, HCC827, or H358 cells compared to those in unstimulated cells. Cells were cultured in the presence of 2 ng/mL TGF-β for 2–5 weeks to induce EMT. Metabolite levels were detected by capillary electrophoresis time-of-flight mass spectrometry. Pathway significance was assessed using $P$ values and false discovery rates (<0.05). The light gray-shaded rows in the table indicate the pathways of amino acid metabolism. **b** and **c** Alteration of amino acid metabolism in A549 cells during TGF-β-induced EMT. **b** Heat map showing the time-course of amino acid levels in A549 cells stimulated with or without 5 ng/mL TGF-β ($n = 4$) for 24, 48, and 72 h. Metabolite levels in each sample were converted to a fold-change relative to average metabolite level of the paired non-stimulation. Red and blue indicate higher and lower levels, respectively, of metabolites in TGF-β-stimulated cells compared to those in the unstimulated cells (white). **c** The metabolites indicated by red or blue on the map for amino acid metabolism were significantly up- or down-regulated by TGF-β treatment for 72 h, respectively.

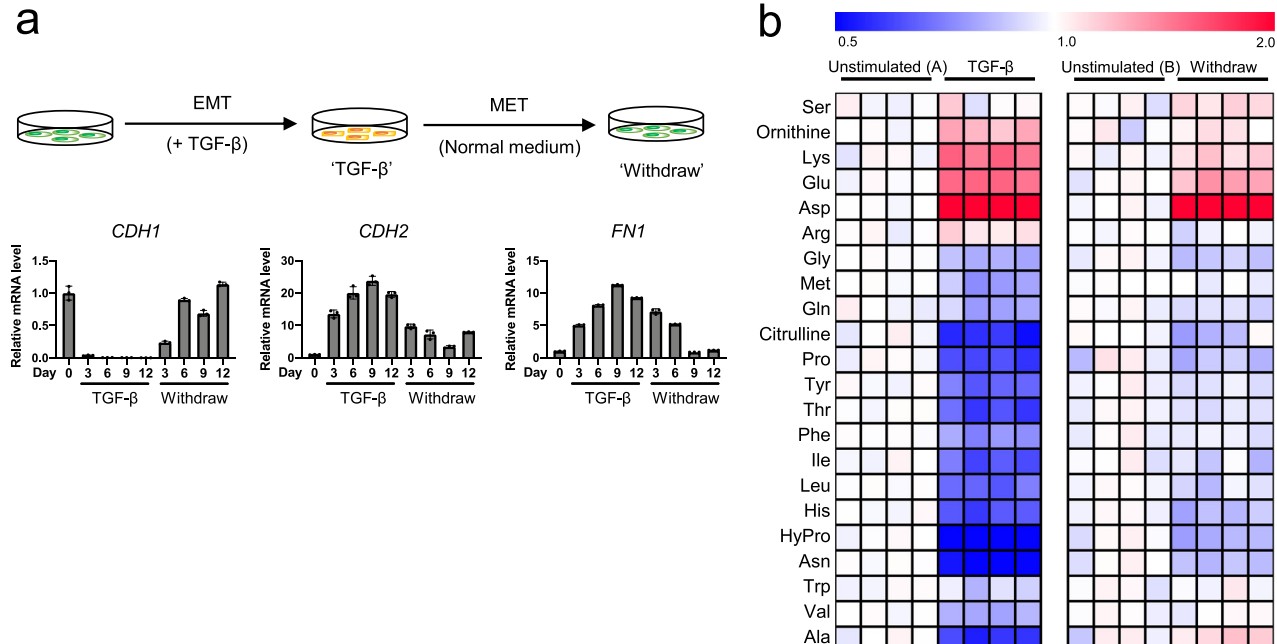

**Fig. 2 Reversible EMT responses induced by TGF-β. a** Reversible EMT marker changes induced by TGF-β in A549 cells. A549 cells were stimulated with TGF-β (2 ng/mL) for the indicated number of days (marked TGF-β). After 12 days, the cells were washed with PBS and subsequently cultured in normal growth medium without TGF-β for 3, 6, or 9 days (marked Withdraw). The cells were passaged for 3 days. mRNA levels of EMT markers *CDH1*, *CDH2*, and *FN1* were measured by real-time PCR. Values are presented as mean ± SD from triplicate samples. **b** TGF-β-induced reversible changes in amino acid levels in A549 cells. 'TGF-β', A549 cells stimulated with TGF-β (5 ng/mL) for 3 days, washed with PBS and subsequently cultured in regular medium without TGF-β for 3 days ('Withdraw'). Controls were cultured in normal growth medium for 3 days ('Unstimulated (A)') and 6 days ('Unstimulated (B)') for TGF-β stimulation and withdrawal, respectively. The medium for 'Unstimulated (B)' was changed to fresh normal medium after culturing for 3 days, as well as for 'withdraw' cultures. Metabolite levels in each sample were converted to a fold-change relative to the average metabolite level of the paired non-stimulation. Red and blue indicate higher and lower levels, respectively, of metabolites in TGF-β-stimulated cells compared to those in the unstimulated cells (white).

and ZEB1 protein levels, in A549 and SW1573 cells (Fig. 3c, d, and Supplementary Fig. 3d). Furthermore, we assessed the effect of amino acids on cell shape and found that depletion of Phe, Thr, Trp, Lys, Val, Met, Leu, Ile, Gln, Arg, or Tyr, but not His, significantly induced morphological changes from a pebble-like shape to an elongated shape in A549 cells (Fig. 3e). Since the viability of A549 and SW1573 cells was strongly reduced by Cys depletion, we could not evaluate the effect of those on EMT phenotype.

We also examined whether the addition of excessive amino acids to regular culture medium abrogated TGF-β-induced EMT in A549 cells. However, the expression of the epithelial marker *CDH1* was not altered by any amino acid in TGF-β-stimulated A549 cells (Supplementary Fig. 3e). These results suggest that the depletion of particular intracellular amino acids is functionally associated with TGF-β-induced EMT in lung cancer cells, and exogenous supplementation of amino acids alone is insufficient to suppress TGF-β-induced EMT.

**Identification of *P4HA3* by integrated metabolomic and transcriptomic analysis.** Next, we sought to explore the mechanism of TGF-β-dependent changes in amino acid metabolism by integrated metabolomic and transcriptomic analyses. The two-layer omics screening method identified four genes: arginase 2 (*ARG2*), alanyl aminopeptidase (*ANPEP*), glutaminase (*GLS*), and prolyl 4-hydroxylase subunit alpha 3 (*P4HA3*), which were upregulated and involved in the alteration of amino acid metabolism in TGF-β-stimulated cells (Fig. 4a and Supplementary Fig. 4a–d). Importantly, GLS inhibitors are previously reported to suppress the EMT traits in lung and breast cancer cells[30,31].

We focused on P4HA3 because its expression is significantly correlated with EMT markers (e.g., *CDH1*, *ZEB1*, and *FN1*) and EMT score (Supplementary Fig. 4e) in 187 lung cancer cell lines from the Cancer Cell Line Encyclopedia (CCLE)[32,33] (Fig. 4b). The level of P4HA3 was also dramatically correlated with mesenchymal status both during the TGF-β stimulation (EMT) and withdrawal (MET) (Fig. 4c, d).

***P4HA3* knockdown abrogates TGF-β-induced amino acid changes and EMT.** We also examined the effect of P4HA3 over-expression on EMT in A549 and HCC827 cells (Supplementary Figs. 5a–d), however, insignificant changes were observed in the EMT markers of P4HA3-overexpressing cells (Supplementary Figs. 5a–d). We then assessed the effect of siRNA-mediated P4HA3 knockdown on EMT marker expression in TGF-β-treated A549 cells. *P4HA3* knockdown increased the expression of *CDH1* and decreased that of *CDH2*, *FN1*, *MMP9*, and *MMP2* (Fig. 5a and Supplementary Fig. 5e). Likewise, in HCC827 and H358 cells, *P4HA3* siRNA abrogated TGF-induced EMT marker patterns (Fig. 5a and Supplementary Fig. 5e). To further examine the role of P4HA3 on an intrinsic EMT status, we used SW1573 cells, which have a mesenchymal character[34] and found that *P4HA3* knockdown caused reversion from the mesenchymal to the epithelial phenotype (Supplementary Fig. 5f).

Next, we examined the role of P4HA3 on the metabolism in TGF-β-stimulated A549 cells. Strikingly, *P4HA3* knockdown abrogated the TGF-β-mediated metabolic changes in amino acid metabolism (Fig. 5b, c, and Supplementary Data 8). In addition, analysis using the metabolome and transcriptome datasets for 147 lung cancer cell lines from the CCLE[32,35] showed that P4HA3 expression and EMT score were significantly correlated with

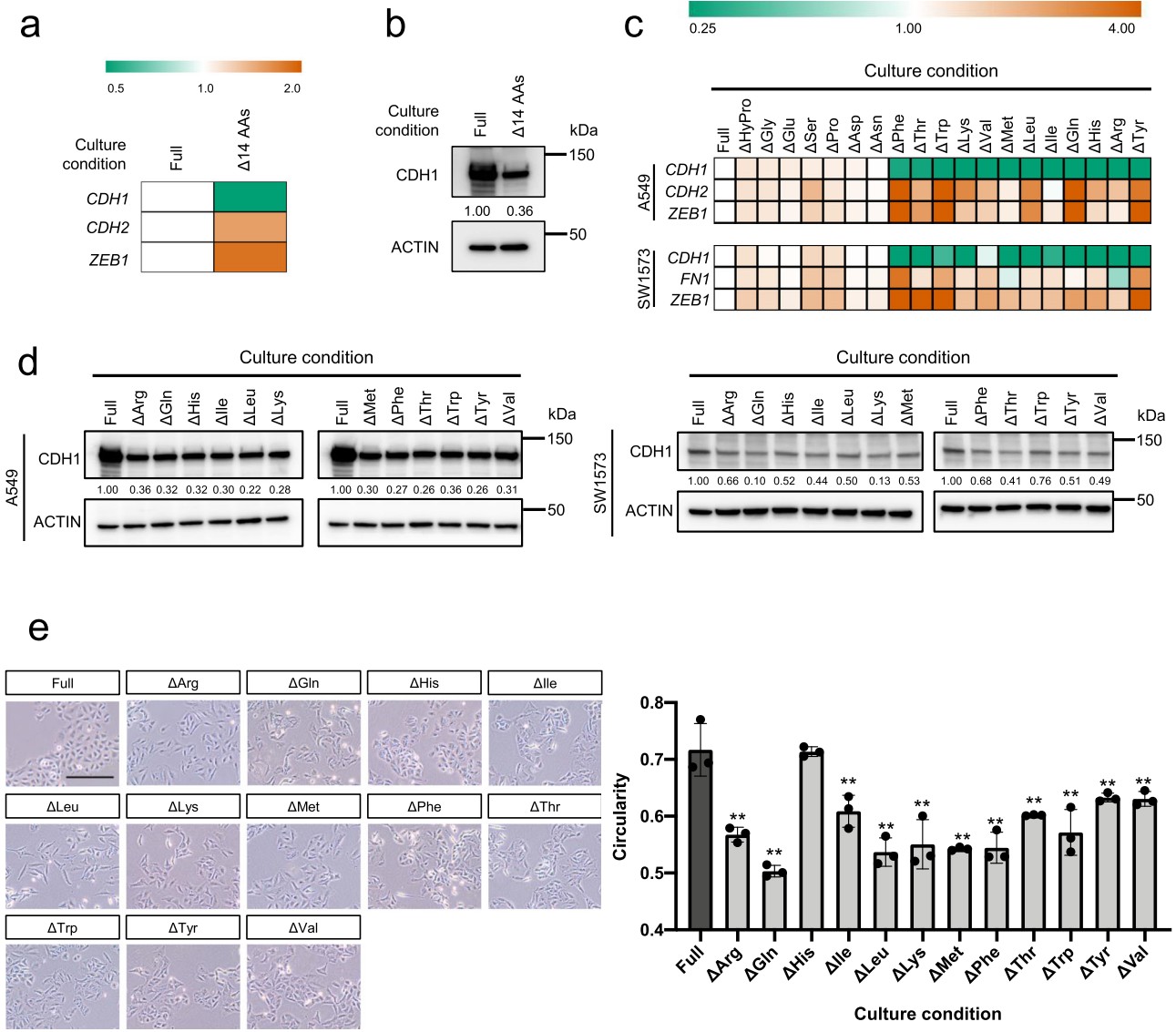

**Fig. 3 Induction of epithelial–mesenchymal transition (EMT) phenotypes by amino acid depletion. a** and **b** Effect of the combined deletion of Met, Val, Pro, Trp, Leu, Ile, Phe, Gly, Tyr, His, Ala, Thr, Asn, and hydroxyproline on **a** mRNA levels of EMT markers and **b** CDH1 protein. mRNA and protein levels were determined by real-time PCR and western blotting, respectively. Orange and green indicate higher and lower levels, respectively, compared to those under regular conditions with full amino acids (white). **c** Effects of amino acid depletion on mRNA expression of EMT marker genes *CDH1*, *CDH2*, *FN1*, and *ZEB1* in A549 and SW1573 cells. Orange and green indicate higher and lower levels, respectively, compared to those under regular conditions with full amino acids (white). **d** Effects of amino acid depletion on CDH1 protein expression in A549 and SW1573 cells. Protein expression levels were measured using western blotting. Actin was the loading control. Relative protein levels of CDH1 quantified using ImageJ software. **e** Effects of amino acid depletion on cellular morphology of A549 cells. Cells were observed by phase-contrast inverted microscope at ×100 magnification. Scale bar, 250 µm. Cell circularity was measured using Image J software. Values are presented as the mean ± SD from triplicate samples. **P < 0.01.

levels of citrulline, Arg, ornithine, His, and Lys (Fig. 5d and Supplementary Fig. 5g). Taken together, these results suggest that P4HA3 expression is essential for the TGF-β-dependent changes in amino acid metabolism, as well as for the transdifferentiation to, and maintenance of, the mesenchymal phenotype; while P4HA3 upregulation alone is insufficient to induce EMT.

**P4HA3 is critical for NSCLC progression and metastasis.** Next, we analyzed the expression of *P4HA3* in NSCLC clinical samples in three independent studies (GSE3141, GSE30219, and GSE31210)[36–38], which revealed a significant positive correlation (*P* < 0.05) between P4HA3 expression and poor prognosis (Fig. 6a). We also investigated P4HA3 expression in tumor tissues

from the lung cancer dataset in the Cancer Genome Atlas (TCGA) and its relationship to TNM staging. P4HA3 expression in tumor tissues (primary and recurrent tumors) was higher than that in non-tumor tissues (Fig. 6b). Interestingly, although there are only two samples for recurrent tumors in this dataset, these samples had high P4HA3 expression. However, the significance of P4HA3 expression in TNM status was not observed (Supplementary Fig. 6a–d).

Next, we investigated the functional role of *P4HA3* in the motility and malignant growth of NSCLC cells. siRNA-mediated knockdown of *P4HA3* significantly suppressed cell migration and growth in vitro (Fig. 6c, d and Supplementary Fig. 6e–h). We also examined the effect of *P4HA3* knockdown on tumor growth and metastasis using a mouse xenograft model. shRNA-mediated

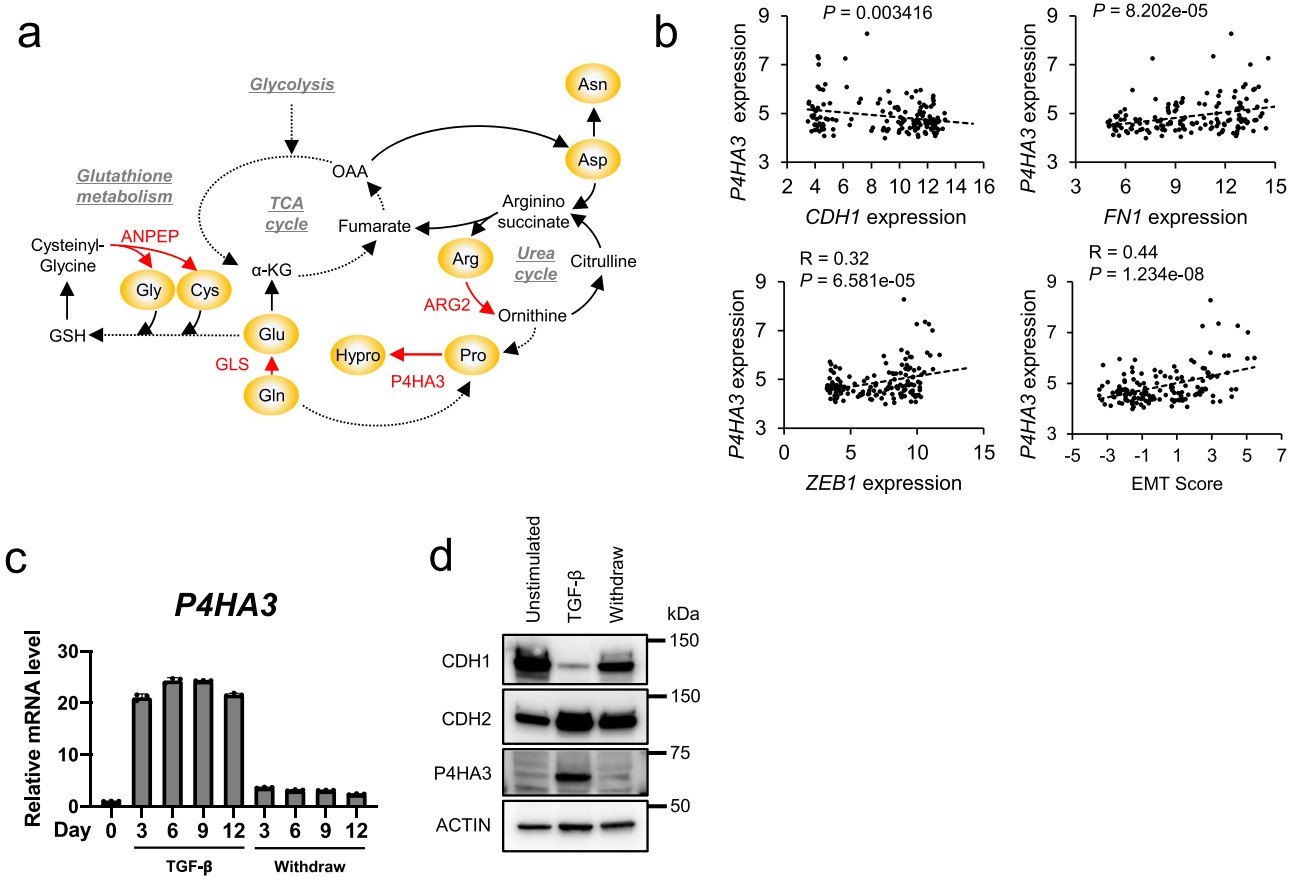

**Fig. 4 Identification of _P4HA3_ by integrated metabolomic and transcriptomic analysis. a** Metabolic reactions of ARG2, ANPEP, GLS, and P4HA3. **b** Correlations between EMT markers and _P4HA3_ mRNA expression in non-small cell lung cancer (NSCLC) cell lines in the Cancer Cell Line Encyclopedia (CCLE) dataset. EMT scores were calculated based on the expression of the reported 76 genes as an EMT marker in NSCLC. **c** Expression of _P4HA3_ during TGF-β-stimulation and withdrawal. A549 cells were stimulated with TGFβ (2 ng/mL) for the indicated number of days (marked TGF-β), and mRNA level of _P4HA3_ was measured by real-time PCR. Values are presented as the mean ± SD from triplicate samples. **d** Protein levels of EMT markers (CDH1 and CDH2) and P4HA3 in A549 cells. A549 cells were stimulated with TGF-β (5 ng/mL) for 3 days (marked TGF-β) and then cultured in normal growth medium without TGF-β for 3 days (marked Withdraw). Actin was used as a loading control for western blot analysis.

_P4HA3_ knockdown significantly reduced the tumor volume in subcutaneous xenografts (Fig. 7a and Supplementary Fig. 7). Moreover, the expression of _CDH1_ in the _P4HA3_ knockdown tumors was higher than that in control tumors (Fig. 7b) and the levels of amino acids in tumors were drastically altered by _P4HA3_ knockdown (Fig. 7c and Supplementary Data 9). Notably, down-regulation of hydroxyproline, Gly, and Trp, as well as the upregulation of ornithine, were statistically significant (Fig. 7c, d). Furthermore, we assessed the effect of _P4HA3_ knockdown on cancer metastasis using a tail vein injection model. _P4HA3_ knockdown markedly suppressed lung metastasis in this model (Fig. 7e). These results suggest that P4HA3 plays an important role in EMT and consequently promotes tumor growth and metastasis in vitro and in vivo.

## Discussion

We identified unique alterations in amino acid metabolism during TGF-β-induced EMT in NSCLC cells. P4HA3 was found to be the key enzyme involved in these metabolic changes, and its expression contributed to EMT phenotypes, such as cell migration (in vitro) and metastasis (in vivo). These findings suggest that the EMT-related metabolic reprogramming of amino acids plays an important role in cancer progression.

The _P4HA3_ gene encodes prolyl 4-hydroxylase subunit alpha 3, a metabolic enzyme that converts proline to hydroxyproline. In

human tissues, three collagen prolyl 4-hydroxylase alpha isoforms (P4HA1, 2, and 3) have been identified. _P4HA1_ and _P4HA2_ expression has been reported to be altered in diverse tumor types, including oral cavity squamous cell carcinoma, prostate cancer, hepatoma, and breast cancer, and involved in tumor progression[39]. P4HA3 has been suggested to be associated with pulmonary fibrosis, which is a mesenchymal cell-associated disease[40]. P4HA3 enhances collagen accumulation through hydroxyproline production, and a P4HA3 inhibitor suppresses hydroxyproline content and fibrotic histopathology in vivo. Recent studies reported that P4HA3 contributes to poor prognosis in gastric or breast cancer[41,42]. In addition, EMT induces expression of angiogenic factors including MMPs[43], suggesting that P4HA3 contributes to angiogenesis through EMT and, consequently, suppresses in vivo tumor growth. Moreover, _P4HA3_ is transcriptionally silenced by DNA methylation in lymphoma[44]. Thus, considering that many genes associated with cellular plasticity in EMT/MET are regulated by epigenetic modifications, including DNA methylation[45], TGF-β might induce these alterations for _P4HA3_.

We also observed that _P4HA3_ knockdown suppressed EMT-induced changes in amino acid metabolism. Intriguingly, even though P4HA3 is the only enzyme involved in proline metabolism, its knockdown extensively abrogated TGF-β-altered amino acid metabolism (Fig. 5c). We also demonstrated that the depletion of specific amino acids from the culture medium induced an

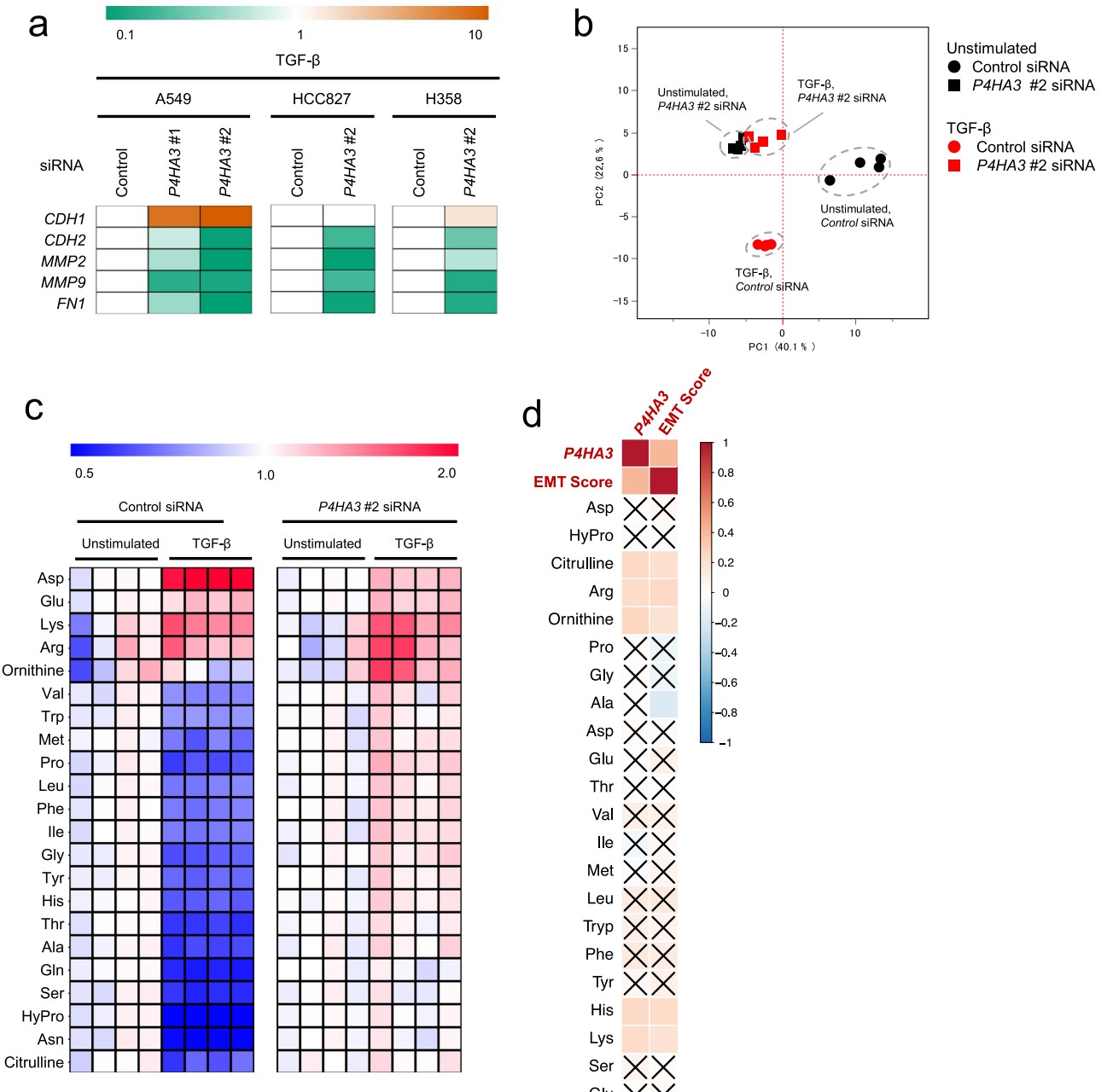

**Fig. 5 Effect of P4HA3 knockdown on epithelial–mesenchymal transition (EMT) marker expression and altered amino acid metabolism in TGF-β-stimulated lung cancer cells. a** Effect of *P4HA3* siRNA on EMT marker expression in TGF-β-treated A549, HCC827, and H358 cells. Cells transfected with P4HA3 siRNAs were treated with TGF-β for 48 h. mRNA expression levels of *CDH1*, *CDH2*, *FN1*, *MMP2*, and *MMP9* were measured by real-time PCR. Orange and green indicate higher and lower levels, respectively, compared to mRNA levels in cells transfected with control siRNA (white). **b** Principal component analysis for metabolomics profiles of the P4HA3-knockdown A549 cells stimulated with or without TGF-β ($n = 4$). Metabolite levels were detected by capillary electrophoresis time-of-flight mass spectrometry. The black and red plots indicate unstimulated and TGF-β-stimulated, respectively, cells. The circle and square indicate control and *P4HA3* siRNA-treated cells, respectively. **c** Effect of *P4HA3* siRNA on TGF-β-induced amino acids alteration in A549 cells. Heat map depicts the ratios of measured sample to mean of unstimulated sample concentrations. Red and blue indicate higher and lower levels, respectively, of metabolites in TGF-β-stimulated cells compared to those in the unstimulated cells (white). **d** Correlations between amino acid levels, EMT score, and *P4HA3* mRNA expression in NSCLC cell lines in the CCLE dataset. Positive correlation is indicated in red and negative correlation in blue. × indicates no significance ($P > 0.05$).

EMT-like phenomenon (Fig. 3). Metabolites in the urea cycle (Arg, ornithine, and citrulline) were commonly identified in the analyses from multiple perspectives for EMT metabolism, including the metabolic changes in TGF-β induced EMT, the amino acid depletion related to EMT induction, and the metabolic alterations in *P4HA3* knockdown tumors (Figs. 1b, 1c, 3c, 3d, 3e,

5c, 5d, 7c). These results strongly suggest metabolic reprograming of the urea cycle during EMT. TGF-β might promote amino acid catabolism by activating urea cycle metabolism, consequently decreasing various intracellular amino acids level. The depletion of Arg from cell culture medium also induced EMT-like phenotypes (Fig. 3c–e). Considering that the proline metabolic pathway,

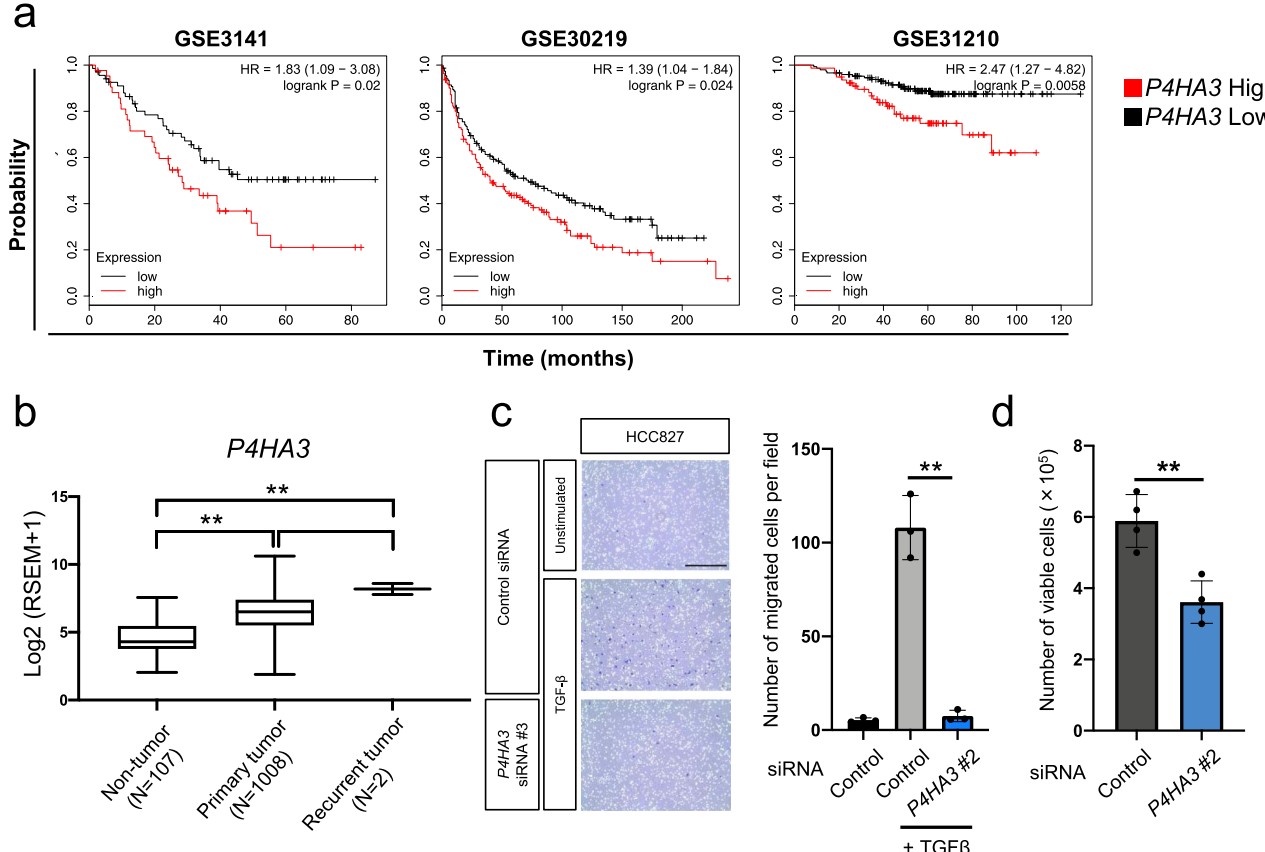

**Fig. 6 Effect of P4HA3 depletion on cell migration and growth in vitro. a** Association between P4HA3 and prognosis in patients with non-small cell lung cancer. **b** Box-and-whisker plot of *P4HA3* mRNA levels in non-tumor ($n = 107$), primary tumor ($n = 1008$), and recurrent tumor tissues ($n = 2$) from the lung cancer dataset in the Cancer Genome Atlas. **P < 0.01. **c** Effects of *P4HA3* siRNA on the migratory activity in TGF-β-treated HCC827 cells. HCC827 cells were cultured in the presence of 2 ng/mL TGF-β for 2–5 weeks to induce EMT and transfected with *P4HA3* siRNA for 48 h prior to the cell migration assay. Cells were observed by bright field microscopy at ×100 magnification. Scale bar, 500 μm. Data are presented as mean ± SD. **P < 0.01. **d** Effect of *P4HA3* knockdown on cell viability in A549 cells. Cells were enumerated 72 h after transfection with *P4HA3* siRNA. Data are presented as mean ± SD. **P < 0.01.

including P4HA3, engages in crosstalk with the urea cycle,[46,47] and that *P4HA3* knockdown prevented the amino acid changes by TGF-β (Fig. 5c), P4HA3 might be indirectly involved in the metabolic activity of the urea cycle. Further, a possible explanation for why an altered EMT status was not observed following depletion of Pro or hydroxyproline (Fig. 3c) may relate to the role of P4HA3 in urea cycle control during EMT induction; alternatively, the cellular abundance of Pro and hydroxyproline may not be associated with EMT, or they may be sufficiently supplemented intracellularly via intrinsic production. Meanwhile, Arginase 1, an enzyme involved in the urea cycle, promotes EMT and tumor metastasis in hepatocellular carcinoma[48], and ARG2, a mitochondrial arginase, was increased by TGF-β in this study (Supplementary Fig. 4c, d). These findings support our hypothesis that the urea cycle contributes to EMT.

Intracellular levels of Lys were also increased by TGF-β (Fig. 1b) and were correlated with EMT scores in the CCLE dataset (Fig. 5d and Supplementary Fig. 5g), while Lys depletion from cell culture medium induced EMT (Fig. 3c–e). Hence, limited cellular availability of Lys might cause intracellular Lys accumulation and contribute to EMT progression. However, further studies are warranted to reveal whether P4HA3 upregulation is epigenetically regulated during EMT, whether P4HA3 rewires amino acid metabolism via regulation of the urea cycle, and how TGF-β-induced loss of amino acids contributes to EMT in cancer cells.

In summary, we demonstrated that EMT inducers, such as TGF-β, alter amino acid metabolism by inducing P4HA3 expression, which might consequently contribute to cancer cell malignant properties in vitro and in vivo. Our results suggest that amino acid metabolism might be a potential therapeutic target for the prevention of metastasis and tumor progression. Additionally, a P4HA3 inhibitor may be a potential anti-cancer agent.

## Methods

**Reagents and cell culture**. Recombinant human TGF-β was purchased from R&D Systems Inc. (Minneapolis, MN, USA). The lung cancer cell lines, A549, HCC827, H358, and SW1573 were obtained from the American Type Culture Collection (ATCC) (Manassas, VA, USA). All cells were grown in RPMI 1640 medium (FUJIFILM Wako Pure Chemical Industries, Osaka, Japan) containing 10% (v/v) fetal bovine serum (FBS), 100 U/mL penicillin, 100 mg/mL streptomycin, and 0.25 mg/mL amphotericin B at 37 °C in a humidified atmosphere with 5% $CO_2$.

**Amino acid depletion medium**. All amino acid-deprived and 10% concentrated RPMI-1640 medium was purchased from Cell Science and Technology Institute Inc. (Sendai, Japan). The 20 amino acids that are included in complete RPMI-1640 were obtained from Sigma-Aldrich (St. Louis, MO) or FUJIFILM Wako Pure Chemical Industries (Supplementary Data 10). In accordance with the concentration of each amino acid in complete RPMI-1640 medium, each amino acid-depletion media, or combined (Met, Val, Pro, Trp, Leu, Ile, Phe, Gly, Tyr, His, Ala, Thr, Asn, and hydroxyproline) depletion medium were prepared by adding the appropriate amino acid combination to the amino acid-deprived medium. Subsequently, each media was supplemented with 10% (v/v) FBS, 100 U/mL penicillin, 100 mg/mL streptomycin, and 0.25 mg/mL amphotericin B.

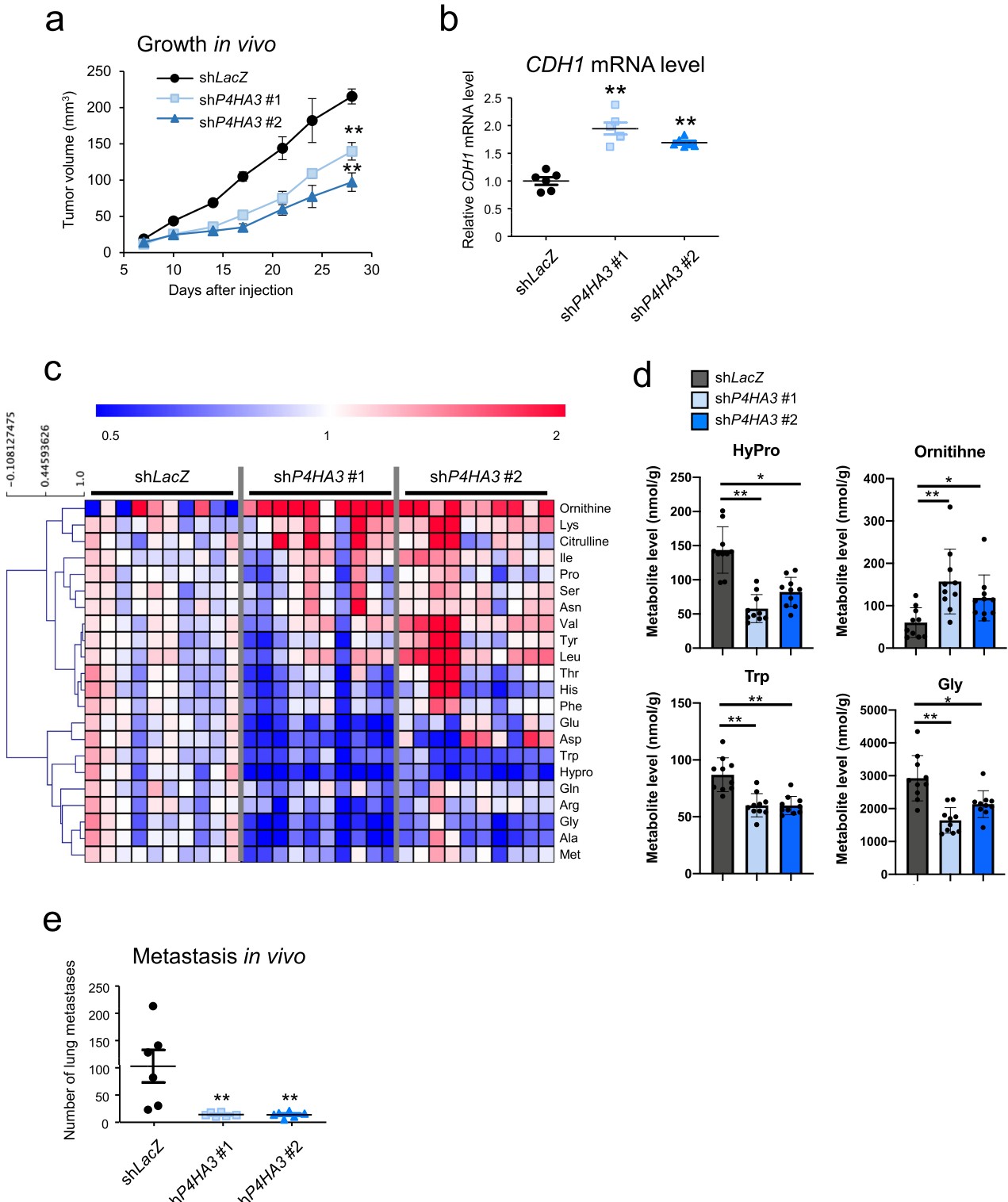

**Fig. 7 Effect of P4HA3 depletion on tumor growth and metastasis in vivo. a** Rate of growth following subcutaneous implantation of control and P4HA3-knockdown A549 cells; $n = 6$ per group. Data represent mean ± SEM. **$P < 0.01$. **b** CDH1 mRNA levels in tumors derived from shLacZ, shP4HA3 #1, and shP4HA3 #2 A549 cells. **c** Altered amino acids in tumors derived from P4HA3-knockdown A549 cells ($n = 10$ per group). Metabolite levels in each sample were converted to a fold-change relative to the average metabolite level of the paired control (shLacZ). Red and blue indicate higher and lower levels, respectively, of metabolites in the P4HA3-knockdown tumors (shP4HA3 #1 and shP4HA3 #2) compared to those in the control tumors (shLacZ) (white). **d** Levels of hydroxyproline, Gly, Trp, and ornithine in the P4HA3 knockdown tumors. Data are presented as mean ± SEM. *$P < 0.05$, **$P < 0.01$. **e** Lung metastatic burden of i.v. injected control and P4HA3-depleted A549 cells 56 days after inoculation. Lung metastatic foci were counted; $n = 6$ per group. Data are presented as mean ± SEM. **$P < 0.01$.

**Determination of morphological changes**. Cells were observed with an EVOS microscope (Thermo Fisher Scientific, Waltham, MA, USA) at ×100 magnification and photographed to evaluate morphological changes. Cell morphological changes were quantitatively determined by calculating cell circularity using NIH ImageJ software (National Institutes of Health, Bethesda, MD, USA). A circularity value of 1.0 indicates a perfect circle. As the value approaches 0, it indicates an increasingly elongated shape. For each condition, the circularity of 90 cells in three separate areas was determined.

**Real-time PCR analysis**. Real-time PCR analysis was conducted as previously described[49]. Briefly, RNA was extracted from cells using TRIzol reagent (Thermo Fisher Scientific) according to the manufacturer's protocols, and 1 μg of RNA was reverse transcribed using the First Strand cDNA Synthesis kit (ReverTra Ace α; Toyobo Co., Ltd., Osaka, Japan). Quantitative real-time PCR was performed using SYBR premix Ex Taq (Takara, Shiga, Japan) on a StepOne Plus Real-Time PCR system (Applied Biosystems, Foster City, CA, USA) according to the manufacturers' instructions. Quantification was performed using the $2^{-\Delta\Delta Ct}$ method, with *RPL27* expression used as an internal reference. Melt curve analysis confirmed that all real-time PCR products were produced as a single DNA duplex. All experiments were performed in triplicate. The primers used for real-time PCR are shown in Supplementary Data 11.

**Western blotting**. Proteins were extracted from the cell lines using RIPA buffer (Sigma-Aldrich) or M-PER reagent (Thermo Fisher Scientific) containing protease inhibitor cocktails (Roche Diagnostics, Indianapolis, IN, USA). Next, protein concentrations were measured by performing a Bradford protein assay. Equal amounts of protein were loaded in each well and resolved with 4–15% Mini-PROTEAN TGX Precast Gels (Bio-Rad Laboratories, Inc., Hercules, CA, USA) at 100 V for 70 min; proteins were subsequently transferred onto a polyvinylidene difluoride membrane (Bio-Rad Laboratories, Inc.) using the Trans-Blot® Turbo™ Transfer System. The membrane was blocked for 3 h with Blocking One reagent (Nacalai Tesque Inc., Kyoto, Japan) with shaking at room temperature. The membrane was then incubated with primary antibody diluted in Can Get Signal® Immunoreaction Enhancer Solution 1 (Toyobo Co., Ltd.) overnight at 4 °C. The membrane was washed three times in PBS with 0.1% tween 20 and incubated with secondary antibody diluted at 1:5000 for 1 h with shaking at room temperature. Next, the membrane was again washed three times for 5 min each. Results were generated using ECL Western Blotting Detection Reagents (GE Healthcare Bio-Sciences, Piscataway, NJ). Immunoreactive bands were visualized using a Luminescent Image Analyzer (LAS-4000 mini; FUJIFILM, Tokyo, Japan). Primary antibodies against ACTIN (1:5000; Santa Cruz Biotechnology, Inc., sc-47778), P4HA3(1:1000; Proteintech, 23185-1-AP), CDH1 (1:1000; Cell Signaling Technology, Denvers, MA, USA, #3195), CDH2 (1:1000; Cell Signaling Technology, #14215), mCherry (1:1000; Abcam, ab167453), GAPDH (1:5000; Ambion, AM4300), and horseradish peroxidase-conjugated secondary antibodies (GE Healthcare Bio-Sciences) were used.

**Metabolite quantification in cells using CE-TOFMS**. Intracellular metabolites were measured by CE-TOFMS (Agilent Technologies, Palo Alto, CA, USA) as previously described[24,50,51]. In brief, cells were washed twice with 5% (w/v) mannitol (FUJIFILM Wako Pure Chemical Industries) and dissolved in 600 μL of methanol containing internal standards (25 μM each of methionine sulfone, ethane sulfonic acid, and d-Camphor-10-sulfonic acid). The homogenate was mixed with 200 μL of Milli-Q water and 400 μL of chloroform. After centrifugation, the separated methanol–water layer was ultra-filtrated through a Millipore 5-kDa cutoff filter (Millipore, Bedford, MA, USA) to remove proteins. The filtrate was lyophilized, dissolved in 25 μL of Milli-Q water containing internal standards (200 μM each of 1,3,5-benzenetricarboxylic acid and 3-aminopyrrolidine), and analyzed using CE-TOFMS. The raw data were processed with MasterHands[52]. Metabolite identities were assigned by matching their *m/z* values and migration times with those of standard compounds.

**Pathway analysis**. The metabolites levels significantly altered by TGF-β ($P < 0.05$, with a false discovery rate [FDR] of 0.05) were analyzed for pathway enrichment using MetaCore[53]. The pathway maps in MetaCore are defined as metabolic cascades that have been experimentally validated and widely accepted. Metabolite identifiers (Kyoto Encyclopedia of Genes and Genomes [KEGG] ID and compound name) were used for each metabolite. The *P* value from the hypergeometric test, generated by MetaCore, represented the enrichment of a certain metabolite in a pathway and was indicative of significant enrichment. The ratio of altered metabolites in the pathway to the total number of metabolites in a pathway was also calculated. Additionally, an FDR of 0.05 was applied for pathway enrichment.

**PCA and heat map visualization**. PCA and heat map visualization were performed as previously described[54]. PCA, a type of unsupervised statistical analysis used widely as a statistical tool in metabolomics studies, was applied prior to detailed data analysis[55]. The metabolites levels were also visualized as a heat map representation. JMP version 12.0.1 (SAS Institute Inc., Cary, NC, USA) and Mev

TM4 software (version 4.7.4. Dana-Farber Cancer Institute, Boston, MA, USA) were used for PCA and heat map analysis, respectively.

**Microarray analysis**. Total RNA was isolated from A549 cells treated with or without TGF-β using the RNeasy Mini Kit (Qiagen, Venlo, Netherlands). The quality of the RNA was assessed using the Agilent 2100 Bioanalyzer (Agilent Technologies). The cRNA amplified from 100 ng total RNA was labeled using Low Input Quick Amp Labeling Kit, One-Color (Agilent Technologies), hybridized to the SurePrint G3 Human GE 8x60K v2 microarray (Agilent Technologies), and scanned using an Agilent scanner according to the manufacturer's instructions. Relative hybridization intensity and background hybridization values were calculated using the Agilent Feature Extraction Software (Agilent Technologies). The data from expression microarray analysis were analyzed using the GeneSpring software (Agilent Technologies).

**RNA interference**. P4HA3 and negative control siRNA duplexes were purchased from Sigma-Aldrich (negative control, SIC-001; P4HA3#1, SASI_Hs01_00206435; P4HA3#2, SASI_Hs01_00206436). Cells were seeded in six-well plates overnight and then transfected with 100 pmol of siRNA oligomer mixed with Lipofectamine RNAiMAX reagent (Thermo Fisher Scientific) in serum-reduced Opti-MEM (Thermo Fisher Scientific) according to the manufacturer's instructions.

**Cell viability assay**. Cell viability was measured by 3-(4,5-dimethylthiazol-2-yl)-2,5-diphenyltetrazolium bromide (MTT, Sigma-Aldrich) assay or trypan blue exclusion assay. MTT assay was performed as previously described[56]. Briefly, cells ($2.5 \times 10^3$ cells/well) were seeded in each well of a 96-well plate and incubated for 24 h. The cells were cultured in amino acid-depletion media for an additional 72 h. Thereafter, 50 μL of MTT (2 mg/mL in PBS) was added to each well and the plates were incubated for an additional 2 h. The resulting formazan crystals were dissolved in 100 μL of dimethyl sulfoxide (DMSO) after aspiration of the culture medium. Plates were placed on a plate shaker for 1 min and read immediately at 570 nm using a TECAN micro-plate reader with Magellan software (Tecan Group Ltd., Männedorf, Switzerland).

The trypan blue exclusion assay was performed by seeding $1 \times 10^5$ cells/well into 12-well cell culture plates and incubating them for 24 h at 37 °C. Seventy-two hours after P4HA3 siRNA transfection, cells were disaggregated in 500 μL medium and 10 μL of the suspension was mixed with 10 μL trypan blue (Thermo Fisher Scientific). Viable cell counts were obtained using the Countess Automated Cell Counter (Thermo Fisher Scientific).

**Cell counting using DAPI staining**. Cells were fixed using 4% paraformaldehyde for 15 min and subsequently permeabilized with 0.1% Triton-X100 for 5 min. Nuclei were stained with 1 μg/mL 4′,6-diamidino-2-phenylindole (DAPI; Nacalai Tesque, Inc.). Fluorescent images were obtained using the high-content imaging device IN Cell Analyzer 2500HS (GE Healthcare Life Science, Chicago, IL, USA) with a ×20 objective lens. Images ($2040 \times 2040$ pixels) from 30 fields were collected for each well.

**Correlation analysis for amino acid level, EMT markers, and P4HA3 expression in lung cancer cell lines**. Metabolome and transcriptome data for 187 lung cancer cell lines were obtained from the CCLE dataset[32,35] (https://portals.broadinstitute.org/ccle). An EMT score for each sample was calculated as the average expression level of mesenchymal genes minus the average expression level of epithelial genes, using the 76 reported EMT signature genes[33]. These data were visualized by the correlation plot package "corrplot" (https://peerj.com/articles/9945/Supplemental_Data_S10.pdf). R package "corrplot": Visualization of a Correlation Matrix (Version 0.84) is available from https://github.com/taiyun/corrplot in the R platform.

**Kaplan–Meier survival analysis**. Survival differences at the gene expression level were validated by KM-Plotter (http://www.kmplot.com/analysis/index.php?p=service&cancer=lung). P4HA3 (228703_at) were entered as the gene symbols, and the checkbox of auto select best cutoff value was selected for *P4HA3* mRNA expression to divide patients into high and low expression groups. Univariate Cox regression was performed to compute the hazard ratio (HR; with 95% confidence intervals) and *P* values. We used multiple GSE datasets (GSE3141, GSE30219, and GSE31210) to evaluate the correlation between the expression of each metabolic gene and patient outcome.

**The Cancer Genome Atlas data**. Data for P4HA3 mRNA expression and TNM staging from "TCGA Lung Cancer (LUNG)" cohort were downloaded from UCSC Xena (https://xena.ucsc.edu). The box-and-whisker plots in Fig. 6b and Supplementary Figs. 6a–d were created using GraphPad Prism software (GraphPad Software, Inc., La Jolla, CA, USA). Kruskal-Wallis test was used for statistical analysis.

**In vitro cell migration assays**. For transwell migration assays, approximately $5.0 \times 10^4$ cells in 100 μL of serum-free RPMI 1640 were seeded onto filter inserts (Corning Transwell culture inserts; Corning Costar, Cambridge, MA, USA). Medium containing 10% FBS was placed in the lower chamber as a chemoattractant to encourage cell migration. After 4 h of incubation, inserts were fixed and stained with a Diff-Quik kit (Sysmex Corporation, Hyogo, Japan). The non-migrated cells were removed using cotton swabs and air-dried. Membranes were photographed using an EVOS microscope at ×100 magnification. We selected three random views per membrane to count the number of cells. Each independent experiment was repeated three times. Data are presented as the number of migrated cells per field.

**Vector construction**. Control shRNA targeting LacZ had the following sequence: 5′-gcuacacaaaucagcgauuucgaaaaaucgcugauuuuguguagc-3′. shRNA against P4HA3 had the following sequences: 5′-gggauuauuaccaugccauuccgaagaauggcaugguaauaauccc-3′ (#1) and 5′-ggauggccaggaaugucuugacgaaucaagacauuccuggccaucc-3′ (#2). Targeted gene sequences were subcloned as deoxyribose fragments into pENTR/U6 TOPO (Thermo Fischer Scientific) and recombined into the lentivirus vector pLenti6 BLOCKiT. Lentiviral vectors were generated and used according to the manufacturer's instructions.

PiggyBac tetO-P4HA3-ires-mCherry-rEF1a-rtTA vector: Vector construction was conducted as previously described[57]. Briefly, P4HA3 cDNA was synthesized by FASMAC (Kanagawa, Japan). The cDNA fragment (1635 bp) of *P4HA3* was cloned using PrimeSTAR® Max DNA Polymerase (Takara). The Kozak sequence was added to the 5′ end of *P4HA3* cDNA. This fragment was inserted in pCR8-GW-TOPO using Topo cloning technology (Thermo Fisher Scientific) and inserted into PiggyBac (PB) tetO-attR1-ccdB-attR2-ires-mCherry-rEF1a-rtTA vector using Gateway technology. Cells were transfected with this plasmid and pCAG-PBase using Lipofectamine 2000 (Thermo Fisher Scientific) and selected with 1.4 mg/mL G418 (Nacalai Tesque, Inc.).

**Tumor transplantation**. Experimental protocols were approved by the Animal Care and Use Committees of The Institute of Medical Science, The University of Tokyo. The tumorigenicity of cells was examined using 6-week-old female BALB/c nude mice (Clea Japan, Tokyo, Japan). For the tumor growth assay, $1 \times 10^6$ A549 cells were injected subcutaneously into the dorsal side of the mice. Subsequently, the implanted tumors were blindly measured using a caliper on the indicated days and their volumes were calculated using the formula $V = (L \times W^2)/2$, where $V$ is the volume ($mm^3$), $L$ is the largest tumor diameter (mm), and $W$ is the smallest tumor diameter (mm). In the metabolic analysis in Fig. 7c, the excised tissues were immediately frozen in liquid nitrogen and stored at −80 °C until use. Extraction and quantification of metabolites using CE-MS were performed as described previously[58].

For the lung metastasis assay, mice were injected with $1 \times 10^6$ A549 cells in 0.2 mL PBS via the lateral tail vein and sacrificed 56 days after injection. Lung metastatic foci were counted under a stereoscopic microscope.

**Statistics and reproducibility**. We performed all experiments at least twice and confirmed similar results. Statistical analyses were performed using Microsoft Excel 2016 for Mac (Microsoft, Seattle, WA, USA), GraphPad Prism software (GraphPad Software, Inc.) and R[59]. In in vitro experiments, data for two groups and more than two groups were analyzed using the Student's $t$ test and the one-way analysis of variance, respectively. In mouse experiments, the Kruskal–Wallis test was used for statistical evaluations. The correlations between amino acid level, EMT marker, and P4HA3 expression in Figs. 4b and 5d were analyzed using Pearson's correlation. Data are represented as mean ± SEM or ±SD; $P$ values < 0.05 were considered significant.

**Reporting summary**. Further information on research design is available in the Nature Research Reporting Summary linked to this article.

## Data availability
The data generated or analyzed during this study are provided in the article and supplementary files. Metabolome data are included in Supplementary Data. Microarray data were deposited in the National Center for Biotechnology Information GEO with the accession code GSE136780. Source data are provided in Supplementary Data 12. Uncropped images of western blots are provided in Supplementary Fig. 8. All other data will be available upon reasonable request.

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

## Acknowledgements
We thank K. Igarashi and R. Hayasaka for technical assistance, and Drs. T. Nishihara and T. Ishikawa for helpful discussions and insights. This study was supported by the Japan Society for the Promotion of Science KAKENHI (16K21361, S.T.; 17KK0199, S.T.; 19J22093, F.N.), the Naito Foundation (S.T.), the Mori Memorial Research fund (F.N.), the Yamagishi Student Project Support Program of Keio University (F.N.), the NIH National Institute of Neurological Disorder and Stroke (R01NS089815, A.T.S.), Extramural Collaborative Research Grant of Cancer Research Institute, Kanazawa University (S.T.), and the research funds from the Yamagata Prefecture Government and City of Tsuruoka, Japan (S.T., A.H., M.S., M.T., and T. Soga).

## Author contributions
Conceptualization, F.N. and S.T.; metabolic analysis, F.N., S.T., A.H., M.O., A.U., M.T., and T. Soga.; bioinformatics, F.N., S.T. and M.S.; mouse experiments, T. Sakamoto, F.N., S.T. K.U., H.E., and H.G.; molecular biology experiments, F.N., S.T., T. Sakamoto, T.Y., and K.U.; ideas and critical comments, F.N., S.T., T. Sakamoto, Y.N., A.T.S., S.Y., and T. Soga; writing, F.N., S.T., A.T.S., and T. Soga.; supervision, Y.N., Y.Y., M.T., A.T.S., S.Y., and T. Soga.

## Competing interests
The authors declare no competing interests.
