## [Peer Review File · Communications Biology]

Reviewers' Comments:

Reviewer #1:

Remarks to the Author:

Dear,

I have now carefully read and examined the manuscript entitled "TGF- β -dependent reprogramming of amino acid metabolism induces epithelial-mesenchymal transition in non-small cell lung cancers" by Nakasuka et al. The authors report a reprogramming of intracellular amino acid metabolism upon TGF- β stimulation, which supports EMT in NSCLC cells. Moreover, they identify prolyl 4-hydroxylase alpha3 (P4HA3) as a key metabolic enzyme to support TGF β -induced metabolic changes and EMT.

Overall, the manuscript is well written and data are well presented. The methodological approaches are well designed and allow to draw robust conclusions.

I have only minor comments:

-line 66: edit text as follows "EMT has been shown to enhance acquisition..."

-the authors must edit the text to refer to "amino acid deprivation" instead of "knock out" (strictly used for genetic manipulation). Moreover, it is unclear how the authors have carried out such experiments. Methodological details must be added to better describe how culture medium has been deprived from specific amino acids.

-introduction: EMT and TGF β signaling have been reported to be associated with changes in fatty acid metabolism. Some references must be added (eg Jiang et al Oncogene 2015; Corbet et al Nat Commun 2020; Dalmau et al Mol Biosyst 2015).

Reviewer #2:

Remarks to the Author:

The authors begin by introducing EMT as an important biological process that drives cancer metastasis and "acquisition of a stem-like phenotype", and link EMT to response to stresses such as ROS and nutrient deprivation. They describe TGF β as a potent inducer of EMT, and it's well defined role in modulating EMT-regulating transcription factors such as SNAI and TWIST. However, they state the role of intracellular metabolism in the induction/maintenance of EMT is "relatively unexplored and poorly understood". They then detail a number of seminal studies in showing how metabolite accumulation/increase can drive EMT, but end with the statement "however, the function and mechanism of metabolites that promote EMT remains largely unknown".

Whilst not a major point, one would suggest that given these seminal papers they rightly point out, the "largely unknown" statement is not truly accurate, and they should place less emphasis on the "unknown role of metabolite changes in driving/maintaining EMT" in setting scene for study. Furthermore, there are a number of papers not discussed, touching on the role of glycolysis, TCA cycle (IDH and SDH are further examples to FH), and redox management. (Review by Hua et al, 2019, "TGF β -induced metabolic reprogramming during epithelial-to-mesenchymal transition in cancer" covers a number of relevant points that may add to both introduction and discussion)

The authors then set out to profile the changes in metabolite abundance induced by TGF β treatment in NSCLC cell lines driven by KRAS/EGFR mutations by mass spectrometry. The methodology of detection and metabolite analysis is sufficiently thorough to give this reviewer confidence in their findings. These analyses demonstrated TGF β altered amino acid metabolism across all lines interrogated. Further interrogation by timecourse analysis demonstrated that the abundance of a number of amino acids are upregulated/downregulated in response to TGF β , and is reversed by removal (except for Asp – this could be discussed further). Is there a significance to which AA's are upregulated vs downregulated? Given the plasticity of this response, does it suggest an epigenetic mechanism may be responsible for what regulates the changes in metabolite abundance? Given the previous literature regarding this, it would be an important point to consider.

To examine the impact of amino acid depletion on EMT induction, the authors then examined CDH1, CDH2 and ZEB1 expression in response to AA depletion. Whilst indeed depletion of AA from media alter CDH1 expression, it is not entirely consistent with the changes observed in response to TGF β . These discrepancies are not discussed and should be expanded upon. Furthermore, it has

been demonstrated that the reduction in amino acid availability impacts viability in cancer cell lines (example: Maddocks et al, 2013; 2017) and therefore viability should be tested in response to AA depletion to uncouple phenotypes. Nonetheless, a change in EMT-associated proteins are observed in response to AA deprivation. It would be nice to determine if indeed there were changes in the associated phenotypes of EMT coupled to this change in protein expression, by methods such as those shown for TGFb in supplementary figure 1).

TGFb is needed to induce EMT, but obviously other factors can influence given that enzyme expression correlates with EMT score – is there a suggestion that the metabolite abundance changes would be influenced in a similar way by alternative mechanisms of EMT signalling?

To determine the mechanism of TGFb-induced metabolic changes, the authors integrated metabolite abundance and transcriptional datasets, identified four genes and focussed on P4HA3 as their lead candidate. They demonstrate that it's expression correlates with TGFb stimulation/withdrawal, EMT scores, and indeed knockdown controlled the response to TGFb. However, this protein is important in regulating Proline metabolism, and the discrepancies between Proline/Hydroxyproline depletion in Figure 2 e&f are not discussed in sufficient detail. Furthermore, to explore a direct link between metabolite abundance and P4HA3 and EMT signatures in this cell line panel, this reviewer would suggest examining datasets published by Chen et al, 2019 across a panel of NSCLC to strengthen/support these findings.

The relevance of P4HA3 is then explored in three clinical NSCLC datasets, and demonstrate that high expression correlates with poor overall survival outcomes (however, this could potentially be expanded by examining grade/TMN status, and is from datasets that involve treated patients, which can confound outcomes. The implications of this should be interrogated and discussed.) P4HA3 knockdown by siRNA decreases cell motility across two cell lines, and however viability of the siRNA-treated cells are not tested, only the stable lines demonstrate "viability, as measured by MTT". Stable line generation is often as a result of selection, here it is a result of blasticidin resistance, therefore only viable cells will grow out. A determination of the impact on viability, using a method appropriate when manipulating metabolic networks, is needed on acute siRNA transfection to determine if the decreased motility is confounded by changes in cell viability. MTT is not an appropriate measure of viability when manipulating metabolism.

The paper ends with a demonstration that knockdown of P4HA3 decreases xenograft tumour growth in a subcutaneous model, and decreased colonisation capability in a tail vein metastasis assay. This is a nice demonstration of the role of this protein in mediating metastatic capacity. A concern with this study is that the impact of amino acid relevance in vitro in lung cancer is debatable, as demonstrated by Davidson et al 2016, so confirmation of metabolite abundance changes, and absence of these with knockdown of P4HA3 would be important – in vivo is possible as evidenced by Figure 4d-f. Furthermore, it would be nice to determine if expression of this gene correlates with other EMT markers in patient samples (Figure 4a), or is it a different function of P4HA3 in vivo that mediates poor outcomes?

In addition, little comment is made to what could be regulating P4HA3 expression/function. Given the links between EMT and methylation, and publications such as Hatzimichael et al, 2012 BJC, demonstrating P4H methylation regulates expression, it would be appropriate to determine if this was altered in response to TGFb – induced EMT.

In summary, the manuscript demonstrates a TGFb-driven EMT alters amino acid metabolism across 3 cell line models of NSCLC – this is comprehensive, and conclusive. It links P4HA3 to such changes, and demonstrates this could be a potential target to inhibit tumour growth and metastasis in xenograft models.

The data is presented well, and with appropriate detail in the methodology to allow replication.

A major concern for this reviewer is that lung tumour metabolism has been demonstrated to be context dependent, and therefore the findings from this study would be strengthened by supporting in vivo metabolomics work, and enhance relevance to the wider community.

With ever increasing knowledge in this field, further efforts could be made to improve discussion of

the relevance of these findings in the wider context of EMT and metabolism.

Reviewer #3:

Remarks to the Author:

In general, the observation of the manuscript entitled "TGF- β -dependent reprogramming of amino acid metabolism induces epithelial-mesenchymal transition in non-small cell lung cancers" by Nakasuka et al, is interesting and some of the ideas are novel. However, the current study is somewhat diffused and there are several issues with the experiments and the conclusion.

Specific comments:

1. In Figure 2, it would be helpful to provide the images of cell morphology for better understanding.
2. In Figure 2C-F, it is unclear how authors knocked out amino acids as group or single. Please provide a brief method. Although, I understand that authors tried to use TGF- β treated media (since a bunch of amino acids were downregulated), however; a. downregulation was a half fold. Is it sufficient? It does not seem to be called "depletion". b. in Figure 2C, this is somewhat unclear. So, authors treated TGF- β to A549 cells and incubated for 3 days to downregulate a bunch of amino acids, and grew A549 cells again with this media? During this second incubation, does this media still contain TGF- β ? c. in Fig. 2E and F, authors used single amino acid knockout (KO) media. So, if authors were able to generate and use single amino acid KO media, why did authors not use group amino acids KO media? It would be a lot easier and clear.
3. In Figure 2B-F, group amino acids KO and single amino acid KO show very similar results. However, the results of Asn and Lys KO (Figure 2E and F) are somewhat opposite in Figure 2B.
4. In Figure 2B, Asp was not changed after the withdrawal of TGF- β on 3 days. Then, what about on 9 or 12 days? Based on Figure 2A, the reversal from EMT was almost completely done on those days.
5. In Figure 2F, what about CDH2 protein level? Since the change of CDH1 protein level does not seem to be very strong (although, mRNA level was significantly changed), it would be helpful to provide CDH2 protein level, too. And also, it would be helpful to provide the images of cell morphology
6. In Supple Figure 2, have authors performed this experiment with all indicated amino acid together, not just with single amino acid individually? A single amino acid may not be sufficient, but as a group, it might show different results. In addition, what about CDH2? Was it decreased or was not changed like CDH1?
7. Authors mentioned, "P4HA3 upregulation is essential for the TGF- β -dependent changes in amino acid metabolism, and transdifferentiation to and maintenance of mesenchymal phenotype." on Page 7, Lane 174-176, but all results were generated by knockdown experiments after treatment of TGF- β . So, how about overexpression of P4HA3 in A549 cells with or without TGF- β ? Could it be sufficient to induce EMT?
8. As shown in Figure 3D, it seems like TGF- β induces P4HA3 significantly. However, Figure 4A shows the correlation only by P4HA3, not TGF- β -induced P4HA3. Also, this is one of the reasons for specific comment No. 7.
9. It seems like Figure 4B and C experiments were performed without TGF- β . However, in Figure 3D, parental A549 cells (without TGF- β) express very weak P4HA3. Was the knockdown of this weak protein sufficient to generate Figure 4 results? It would be helpful to provide P4HA3 protein level after knockdown in A549 cells.
10. In Supple Figure 4D, knockdown of P4HA3 does not have a strong impact on cell proliferation,

but it reduced tumour growth significantly in vivo (Figure 4D). Could you explain this discrepancy?

11. From Figure 1 to Figure 3, all experiments were performed with TGF- β . However, Figure 4 experiments were largely performed without TGF- β . Is there any reason that authors did not use TGF- β in Figure 4 experiments? If yes, what is the rationale?

12. Authors mentioned, "Intriguingly, even though P4HA3 is the only enzyme involved in proline metabolism, its knockdown extensively abrogated TGF- β -altered amino acid metabolism." on Page 8, Lane 213-214, however, I think it was not provided as Figure (although it is mentioned in Supple Figure 3B). Could you show the change of amino acid level after knockdown or overexpression of P4HA3?

13. Authors mentioned, "However, the function and mechanism of metabolites that promote EMT remains largely unknown." on Page 4, Lane 97-98, and this current study still did not show the molecular mechanism. The experiments with P4HA3 are quite interesting, but the relationship between the change of amino acids level by TGF- β and P4HA3 was not described here. (However, I agree with what authors said that "Further studies are necessary to reveal the molecular mechanisms of P4HA3 or proline metabolism that widely regulate amino acid metabolism, and also how TGF β -induced loss of amino acid promotes EMT status in cancer cells.")

14. Figure 1 and 2 describe the change of amino acid level by TGF- β and EMT, and Figure 3 and 4 focus on P4HA3 and EMT. The connection between these groups is somewhat weak, which should be more organised.

Minor points:

1. In Supple Figure 1G-I, PCA result of HCC827 seems to be opposite of A549 and H358, even Supple Figure 1A-F show similar results.

2. In Figure 2B, legend and labelling are a little bit confusing. What does this "Unstimulated" mean in the right panel? I thought the withdrawal was performed after TGF- β treatment. As described in the legend, if this unstimulated is just normal growth media, then how did authors compare with the result after the withdrawal? a. Withdraw: I believe this was done like TGF- β induction for 3 days, and changed with normal media, then incubation for further 3 days. b. Unstimulated: So, this is just normal media for 6 days?

3. In Figure 4F, it is a little bit difficult to see the metastatic foci in the left image. (Not sure whether it is due to the low resolution of the image)

4. On Page 8, Lane 216, authors mentioned "Figure 6". Where are Figure 5 and 6?

Nakasuka *et al.* “**TGF- β -dependent reprogramming of amino acid metabolism induces epithelial–mesenchymal transition in non-small cell lung cancers**” COMMSBIO-20-2240

We thank the Reviewers for their helpful comments related to our manuscript. Our point-by-point responses are provided below, in blue text.

Responses to reviewer #1

*I have now carefully read and examined the manuscript entitled “TGF- β -dependent reprogramming of amino acid metabolism induces epithelial–mesenchymal transition in non-small cell lung cancers” by Nakasuka *et al.* The authors report a reprogramming of intracellular amino acid metabolism upon TGF-beta stimulation, which supports EMT in NSCLC cells. Moreover, they identify prolyl 4-hydroxylase alpha3 (P4HA3) as a key metabolic enzyme to support TGFbeta-induced metabolic changes and EMT.*

Overall, the manuscript is well written and data are well presented. The methodological approaches are well designed and allow to draw robust conclusions.

Response #1.1:

We thank the reviewer for their positive feedback.

-line 66: edit text as follows “EMT has been shown to enhance acquisition...”

Response #1.2:

We appreciate this comment and have revised the text accordingly (page 3, line 23).

-the authors must edit the text to refer to “amino acid deprivation” instead of “knock out” (strictly used for genetic manipulation). Moreover, it is unclear how the authors have carried out such experiments. Methodological details must be added to better describe how culture medium has been deprived from specific amino acids.

Response #1.3:

We appreciate this helpful comment and have revised “knock out” to “depletion” throughout the manuscript. We have also included the methods corresponding to the depletion of amino acids in media (page 10, line 29 - page 11, line 7).

*-introduction: EMT and TGF beta signaling have been reported to be associated with changes in fatty acid metabolism. Some references must be added (eg Jiang *et al* Oncogene 2015; Corbet *et al* Nat Commun 2020; Dalmau *et al* Mol Biosyst 2015).*

Response#1.4:

We agree the reviewer and have added references for fatty acid metabolism in the revised Introduction section (page 4, lines 14–15).

Responses to reviewer #2

The authors begin by introducing EMT as an important biological process that drives cancer metastasis and “acquisition of a stem-like phenotype”, and link EMT to response to stresses such as ROS and nutrient deprivation. They describe TGFb as a potent inducer of EMT, and it’s well defined role in modulating EMT-regulating transcription factors such as SNAI and TWIST. However, they state the role of intracellular metabolism in the induction/maintenance of EMT is “relatively unexplored and poorly understood”. They then detail a number of seminal studies in showing how metabolite accumulation/increase can drive EMT, but end with the statement “however, the function and mechanism of metabolites that promote EMT remains largely unknown”.

Whilst not a major point, one would suggest that given these seminal papers they rightly point out, the “largely unknown” statement is not truly accurate, and they should place less emphasis on the “unknown role of metabolite changes in driving/maintaining EMT” in setting scene for study. Furthermore, there are a number of papers not discussed, touching on the role of glycolysis, TCA cycle (IDH and SDH are further examples to FH), and redox management. (Review by Hua et al, 2019, “TGFβ-induced metabolic reprogramming during epithelial-to-mesenchymal transition in cancer” covers a number of relevant points that may add to both introduction and discussion)

Response#2.1:

We fully agree with the reviewer’s comment and have improved the introduction section while including key references (page 4, lines 14–15).

The authors then set out to profile the changes in metabolite abundance induced by TGFb treatment in NSCLC cell lines driven by KRAS/EGFR mutations by mass spectrometry. The methodology of detection and metabolite analysis is sufficiently thorough to give this reviewer confidence in their findings. These analyses demonstrated TGFb altered amino acid metabolism across all lines interrogated. Further interrogation by timecourse analysis demonstrated that the abundance of a number of amino acids are upregulated/downregulated in response to TGFb, and is reversed by removal (except for Asp – this could be discussed further). Is there a significance to which AA’s are upregulated vs downregulated? Given the plasticity of this response, does it suggest an epigenetic mechanism may be responsible for what regulates the changes in metabolite abundance? Given the previous literature regarding this, it would be an important point to consider.

Response#2.2:

We appreciate these important comments. We have shown that the amino acids impacted by 72 h TGF- β treatment, in Fig. 1c. Red and blue indicate metabolites that are significantly up- and down-regulated by TGF- β , respectively. We have also included a more detailed discussion regarding the changes in amino acid levels by TGF- β being induced through epigenetic regulation, as follows (page 9, lines 12–15):

“Moreover, P4HA3 is transcriptionally silenced by DNA methylation in lymphoma⁴⁴. Thus, considering that many genes associated with cellular plasticity in EMT/MET are regulated by epigenetic modifications, including DNA methylation⁴⁵, TGF β might induce these alterations for P4HA3.”

To examine the impact of amino acid depletion on EMT induction, the authors then examined CDH1, CDH2 and ZEB1 expression in response to AA depletion. Whilst indeed depletion of AA from media alter CDH1 expression, it is not entirely consistent with the changes observed in response to TGF β . These discrepancies are not discussed and should be expanded upon.

Response#2.3:

We thank the Reviewer for this helpful observation. We have focused on the urea cycle and intracellular Lys levels and have included more relevant discussion as follows, (page 9, line 20 – page 10, line 13):

“Metabolites in the urea cycle (Arg, ornithine, and citrulline) were commonly identified in the analyses from multiple perspectives for EMT metabolism, including the metabolic changes in TGF- β induced EMT, the amino acid depletion related to EMT induction, and the metabolic alterations in P4HA3 knockdown tumors (Figs. 1b, 1c, 2e, 2f, 2g, 3g, 4g). These results strongly suggest metabolic reprogramming of the urea cycle during EMT. TGF- β might promote amino acid catabolism by activating urea cycle metabolism, consequently decreasing various intracellular amino acids level. The depletion of Arg from cell culture medium also induced EMT-like phenotypes (Figs. 2e–g). Considering that the proline metabolic pathway, including P4HA3, engages in crosstalk with the urea cycle,^{46, 47} and that P4HA3 knockdown prevented the amino acid changes by TGF- β (Fig. 3g), P4HA3 might be indirectly involved in the metabolic activity of the urea cycle. Further, a possible explanation for why an altered EMT status was not observed following depletion of Pro or hydroxyproline (Fig. 2e) may relate to the role of P4HA3 in urea cycle control during EMT induction; alternatively, the cellular abundance of Pro and hydroxyproline may not be associated

with EMT, or they may be sufficiently supplemented intracellularly via intrinsic production. Meanwhile, Arginase 1, an enzyme involved in the urea cycle, promotes EMT and tumor metastasis in hepatocellular carcinoma⁴⁸, and ARG2, a mitochondrial arginase, was increased by TGF- β in this study (Supplementary Figs. 3c, d). These findings support our hypothesis that the urea cycle contributes to EMT.

Intracellular levels of Lys were also increased by TGF- β (Fig. 1b) and were correlated with EMT scores in the CCLE dataset (Fig. 3h), while Lys depletion from cell culture medium induced EMT (Figs. 2e–g). Hence, limited cellular availability of Lys might cause intracellular Lys accumulation and contribute to EMT progression. However, further studies are warranted to reveal whether P4HA3 upregulation is epigenetically regulated during EMT, whether P4HA3 rewires amino acid metabolism via regulation of the urea cycle, and how TGF- β -induced loss of amino acids contributes to EMT in cancer cells.”

Furthermore, it has been demonstrated that the reduction in amino acid availability impacts viability in cancer cell lines (example: Maddocks et al, 2013; 2017) and therefore viability should be tested in response to AA depletion to uncouple phenotypes. Nonetheless, a change in EMT-associated proteins are observed in response to AA deprivation. It would be nice to determine if indeed there were changes in the associated phenotypes of EMT coupled to this change in protein expression, by methods such as those shown for TGF β in supplementary figure 1).

Response#2.4:

We thank the Reviewer for these suggestions. We have added the data related to cell growth and morphology in the cells cultured under amino acid deprivation for 72 h (Fig. 2g and Supplementary Figs. 2c, d; page 6, lines 10–12; page 6, lines 16–18). We assessed cell growth by MTT assay and by counting the number of DAPI-positive cells. The results of both assays were similar. However, the alterations of EMT markers by amino acid depletion was independent of the cell growth.

TGF β is needed to induce EMT, but obviously other factors can influence given that enzyme expression correlates with EMT score – is there a suggestion that the metabolite abundance changes would be influenced in a similar way by alternative mechanisms of EMT signalling?

Response#2.5:

We thank the reviewer for these observations. We are also interested in the molecular mechanisms for P4HA3 expression in mesenchymal lung cancer cells. As such, we have discussed epigenetic regulation of P4HA3 expression (page 9, lines 12–15) and would like to clarify this point in future studies.

To determine the mechanism of TGFb-induced metabolic changes, the authors integrated metabolite abundance and transcriptional datasets, identified four genes and focussed on P4HA3 as their lead candidate. They demonstrate that it's expression correlates with TGFb stimulation/withdrawl, EMT scores, and indeed knockdown controlled the response to TGFb. However, this protein is important in regulating Proline metabolism, and the discrepancies between Proline/Hydroxyproline depletion in Figure 2 e&f are not discussed in sufficient detail.

Response#2.6:

We appreciate this helpful comment. In response, we have discussed the role of P4HA3 on metabolic changes in EMT (page 9, lines 27–30), as well as potential reasons for the depletion of Pro or hydroxyproline in Fig. 2e not impacting EMT status, as follows (page 9, line 30 – page 10, lines 3):

“Further, a possible explanation for why an altered EMT status was not observed following depletion of Pro or hydroxyproline (Fig. 2e) may relate to the role of P4HA3 in urea cycle control during EMT induction; alternatively, the cellular abundance of Pro and hydroxyproline may not be associated with EMT, or they may be sufficiently supplemented intracellularly via intrinsic production.”

Furthermore, to explore a direct link between metabolite abundance and P4HA3 and EMT signatures in this cell line panel, this reviewer would suggest examining datasets published by Chen et al, 2019 across a panel of NSCLC to strengthen/support these findings.

Response#2.7:

We appreciate this helpful comment. Unfortunately, the dataset published by Chen et al, 2019 (DOI: 10.1016/j.molcel.2019.08.028) was missing some of the amino acid levels that we had wanted to include. Thus, we used, and analyzed, another metabolome dataset (Li et al, 2019; DOI: 10.1038/s41591-019-0404-8) and found that P4HA3 expression and EMT score were significantly correlated with levels of citrulline, Arg, ornithine, His, and Lys. We have added these results in the revised manuscript (Fig. 3h and Supplementary Fig. 3l; page 7, lines 24–27)

The relevance of P4HA3 is then explored in three clinical NSCLC datasets, and demonstrate that high expression correlates with poor overall survival outcomes (however, this could potentially be expanded by examining grade/TMN status, and is from datasets that involve treated patients, which can confound outcomes. The implications of this should be interrogated and discussed.) P4HA3 knockdown by siRNA decreases cell motility across two cell lines, and however viability of the

siRNA-treated cells are not tested, only the stable lines demonstrate “viability, as measured by MTT”. Stable line generation is often as a result of selection, here it is a result of blasticidin resistance, therefore only viable cells will grow out. A determination of the impact on viability, using a method appropriate when manipulating metabolic networks, is needed on acute siRNA transfection to determine if the decreased motility is confounded by changes in cell viability. MTT is not an appropriate measure of viability when manipulating metabolism.

Response#2.8:

We agree with the reviewer’s comments regarding the limitations of our original report. However, we have since investigated P4HA3 expression in tumor tissues of a lung cancer dataset in the Cancer Genome Atlas (TCGA), as well as its relationship to TNM staging. We found that P4HA3 expression in tumor tissues (primary and recurrent tumors) was higher than that in non-tumor tissues (Fig. 4b). Interestingly, although there are only two samples for recurrent tumors in this dataset, they had high P4HA3 expression. Significance of P4HA3 expression in TNM status was not observed (Supplementary Fig. 4f). We have this result in the revised manuscript (page 8, lines 4–10).

We also examined the effect of P4HA3 siRNA on cell growth in A549 cells. The viable cells were counted in trypan blue exclusion assay 72 h after P4HA3 siRNA transfection (See method; (page 14, lines 17– 21). P4HA3 siRNA suppressed cell growth. We have added this result and described it in the Result section (Fig. 4d; page 8, lines 11–13).

Regarding the migration assay (Fig. 4c and Supplementary Figs. 4e–g), we consider that the suppressive effect of *P4HA3* knockdown on cell growth did not impact the results of this migration assay, as the migrated cells were analyzed a short time (4 h) after equalizing the number of cells.

The paper ends with a demonstration that knockdown of P4HA3 decreases xenograft tumour growth in a subcutaneous model, and decreased colonisation capability in a tail vein metastasis assay. This is a nice demonstration of the role of this protein in mediating metastatic capacity. A concern with this study is that the impact of amino acid relevance in vitro in lung cancer is debatable, as demonstrated by Davidson et al 2016, so confirmation of metabolite abundance changes, and absence of these with knockdown of P4HA3 would be important – in vivo is possible as evidenced by Figure 4d-f. Furthermore, it would be nice to determine if expression of this gene correlates with other EMT markers in patient samples (Figure 4a), or is it a different function of P4HA3 in vivo that mediates poor outcomes?

Response#2.9:

We have analyzed the levels of amino acids in the xenografted tumors and observed altered amino acid metabolism in the *P4HA3* knockdown tumors (Figs. 4g, h). As significantly changed amino acids, the levels of hydroxyproline, Gly, and Trp were downregulated by *P4HA3* knockdown. Meanwhile, the expression of ornithine was upregulated. We described this result in the revised manuscript (page 8, lines 16–20).

Meanwhile, we attempted to determine whether *P4HA3* expression is correlated with EMT score in the TCGA dataset; however, we could not detect this effect. We consider that the clinical tumor samples have high heterogeneity and any correlation between *P4HA3* and EMT score may be masked.

In addition, little comment is made to what could be regulating P4HA3 expression/function. Given the links between EMT and methylation, and publications such as Hatzimichael et al, 2012 BJC, demonstrating P4H methylation regulates expression, it would be appropriate to determine if this was altered in response to TGF β – induced EMT.

Response#2.10:

We appreciate this comment and have added this issue, as well as the reference, to the revised Discussion section, as follows (page 9, lines 12–15):

*“Moreover, *P4HA3* is transcriptionally silenced by DNA methylation in lymphoma⁴⁰. Thus, considering that many genes associated with cellular plasticity in EMT/MET are regulated by epigenetic modifications, including DNA methylation⁴¹, TGF β might induce these alterations for *P4HA3*.”*

*In summary, the manuscript demonstrates a TGF β -driven EMT alters amino acid metabolism across 3 cell line models of NSCLC – this is comprehensive, and conclusive. It links *P4HA3* to such changes, and demonstrates this could be a potential target to inhibit tumour growth and metastasis in xenograft models. The data is presented well, and with appropriate detail in the methodology to allow replication.*

A major concern for this reviewer is that lung tumour metabolism has been demonstrated to be context dependent, and therefore the findings from this study would be strengthened by supporting in vivo metabolomics work, and enhance relevance to the wider community. With ever increasing knowledge in this field, further efforts could be made to improve discussion of the relevance of these findings in the wider context of EMT and metabolism.

Response#2.11:

We appreciate the Reviewer's thoughtful consideration of our study and its interpretations. We have improved the manuscript by adding data on amino acid metabolism in tumors *following P4HA3* knockdown. We have also included a discussion on the changes in amino acid metabolism in EMT. Although some unanswered questions remain regarding the rewiring of amino acid metabolism in EMT, we hope to address these in our future work.

Responses to reviewer #3

In general, the observation of the manuscript entitled "TGF- β -dependent reprogramming of amino acid metabolism induces epithelial-mesenchymal transition in non-small cell lung cancers" by Nakasuka et al, is interesting and some of the ideas are novel. However, the current study is somewhat diffused and there are several issues with the experiments and the conclusion.

Response#3.1:

We appreciate these comments.

Specific comments:

1. In Figure 2, it would be helpful to provide the images of cell morphology for better understanding.

Response#3.2:

We have added data for the changes in cell morphology following amino acid depletion (Fig. 2g).

2. In Figure 2C-F, it is unclear how authors knocked out amino acids as group or single. Please provide a brief method. Although, I understand that authors tried to use TGF- β treated media (since a bunch of amino acids were downregulated), however; a. downregulation was a half fold. Is it sufficient? It does not seem to be called "depletion". b. in Figure 2C, this is somewhat unclear. So, authors treated TGF- β to A549 cells and incubated for 3 days to downregulate a bunch of amino acids, and grew A549 cells again with this media? During this second incubation, does this media still contain TGF- β ? c. in Fig. 2E and F, authors used single amino acid knockout (KO) media. So, if authors were able to generate and use single amino acid KO media, why did authors not use group amino acids KO media? It would be a lot easier and clear.

Response#3.3:

We thank the reviewer for raising these questions. I apologize that the original manuscript was not written clearly in this regard. We have added the methodology for depletion of amino acids in the

revised manuscript (page 10, line 29 – page 11, line 7). In Fig. 2c, we used medium with depleted amino acids. We have also revised the text that you referenced in the Results section as follows (page 5, line 28 – page 6, line 1):

“Specifically, to investigate the effect of TGF- β -induced amino acid decline on EMT, we first prepared media depleted of Met, Val, Pro, Trp, Leu, Ile, Phe, Gly, Tyr, His, Ala, Thr, Asn, and hydroxyproline, thus mimicking the mimic the effect elicited by TGF- β (Fig. 1c), and examined the effect that the media on the expression of EMT markers (Figs. 2c, d, and Supplementary Fig. 2b).”

3. In Figure 2B-F, group amino acids KO and single amino acid KO show very similar results. However, the results of Asn and Lys KO (Figure 2E and F) are somewhat opposite in Figure 2B.

Response#3.4:

We appreciate these important comments and have discussed the effect of intracellular Lys levels on EMT in the Discussion section as follows (page 10, lines 7–10):

“Intracellular levels of Lys were also increased by TGF- β (Fig. 1b) and were correlated with EMT scores in the CCLE dataset (Fig. 3h), while Lys depletion from cell culture medium induced EMT (Figs. 3e–g). Hence, limited cellular availability of Lys might cause intracellular Lys accumulation and contribute to EMT progression.”

4. In Figure 2B, Asp was not changed after the withdrawal of TGF- β on 3 days. Then, what about on 9 or 12 days? Based on Figure 2A, the reversal from EMT was almost completely done on those days.

Response#3.5:

We agree with the reviewer’s comment. We are also interested in the effect of TGF- β withdrawal over 3 days on Asp level. Please note, the experimental conditions in Fig. 2b differ slightly from those in Fig. 2a, while Fig. 3d presents results with the same experimental conditions as Fig. 2b. We assessed the expression of EMT markers following TGF- β withdrawal for 3 days after TGF- β treatment for 3 days. Results show that TGF- β withdrawal sufficiently reversed EMT. Most of the amino acids affected by TGF- β were restored following TGF- β withdrawal. However, the Asp upregulation induced by TGF- β treatment was maintained despite TGF- β withdrawal for 3 days, which suggests that different mechanisms are responsible for Asp upregulation compared to that for other amino acids. We would like to clarify this is in our future work.

5. In Figure 2F, what about CDH2 protein level? Since the change of CDH1 protein level does not

seem to be very strong (although, mRNA level was significantly changed), it would be helpful to provide CDH2 protein level, too. And also, it would be helpful to provide the images of cell morphology

Response#3.6:

We appreciate this comment. We have assessed CDH2 and ZEB1 protein level and have included the corresponding data in the revised manuscript (Supplementary Fig. 2b; page 6, lines 3–7). Contrary to our expectations, CDH2 and ZEB1 protein levels were downregulated rather than upregulated like their mRNA levels. Hence, amino acid depletion might be involved in the posttranscriptional or posttranslational regulation for CDH2 and ZEB1. We also examined the cellular morphological changes following amino acid depletion and found that depletion of Phe, Thr, Trp, Lys, Val, Met, Leu, Ile, Gln, Arg, or Tyr, but not His, significantly induced morphological changes from a pebble-like shape to an elongated shape in A549 cells. We have described this result in the revised manuscript (Fig. 2g; page 6, lines 16–18).

6. In Supple Figure 2, have authors performed this experiment with all indicated amino acid together, not just with single amino acid individually? A single amino acid may not be sufficient, but as a group, it might show different results. In addition, what about CDH2? Was it decreased or was not changed like CDH1?

Response#3.7:

We thank the reviewer for this important comment. To further clarify the molecular mechanisms associated with amino acid functioning in EMT, we are currently investigating how amino acid concentrations, combinations, or treatment times impact EMT phenotypes in multiple cell lines and would like to report them in our next paper.

7. Authors mentioned, "P4HA3 upregulation is essential for the TGF- β -dependent changes in amino acid metabolism, and transdifferentiation to and maintenance of mesenchymal phenotype." on Page 7, Lane 174-176, but all results were generated by knockdown experiments after treatment of TGF- β . So, how about overexpression of P4HA3 in A549 cells with or without TGF- β ? Could it be sufficient to induce EMT?

8. As shown in Figure 3D, it seems like TGF- β induces P4HA3 significantly. However, Figure 4A shows the correlation only by P4HA3, not TGF- β -induced P4HA3. Also, this is one of the reasons for specific comment No. 7.

Response#3.8:

We appreciate this important comment. We have examined the effect of P4HA3 overexpression on EMT in A549 cells and HCC827 cells. However, the changes in EMT markers by P4HA3 overexpression were not clear (Supplementary Fig. 3f–i; page 7, lines 12–14). We also examined the effects under TGF- β stimulation for 48 h and did not observe changes in EMT markers. We consider that P4HA3 is essential to EMT; however, P4HA3 upregulation alone is not sufficient to induce EMT.

9. It seems like Figure 4B and C experiments were performed without TGF- β . However, in Figure 3D, parental A549 cells (without TGF- β) express very weak P4HA3. Was the knockdown of this weak protein sufficient to generate Figure 4 results? It would be helpful to provide P4HA3 protein level after knockdown in A549 cells.

Response#3.9:

We have added the data for endogenous P4HA3 protein level to Supplementary Fig. 4h.

10. In Supple Figure 4D, knockdown of P4HA3 does not have a strong impact on cell proliferation, but it reduced tumour growth significantly in vivo (Figure 4D). Could you explain this discrepancy?

Response#3.10:

We thank the review for this comment. EMT induced expression of angiogenic factors, including MMP2 and MMP9 (Saray et al, 2019; DOI: 10.3389/fonc.2019.01370), suggesting that P4HA3 contributes to angiogenesis through EMT. We have discussed the above effects of P4HA3 on tumor growth *in vivo* in the revised manuscript (page 9, lines 9–12).

On the other hand, following the comment of reviewer#2.8, we have rechecked cell growth in the P4HA3 knockdown cells by enumerating viable cells, but not MTT assay. siRNA-mediated P4HA3 knockdown suppressed cell growth (Fig. 4d; page 8, lines 11–13).

11. From Figure 1 to Figure 3, all experiments were performed with TGF- β . However, Figure 4 experiments were largely performed without TGF- β . Is there any reason that authors did not use TGF- β in Figure 4 experiments? If yes, what is the rationale?

Response#3.11:

Yes, it is. In supplementary Fig. 3k, we show the suppressive effect of P4HA3 knockdown on CDH1 expression in SW1573 cells which have a mesenchymal character. Also, in the analysis using the

CCL4 dataset, P4HA3 expression was correlated with EMT score (Fig. 3b). These results suggest that P4HA3 is important for not only the TGF- β -induced EMT, but also intrinsic EMT that is induced by various cell signaling pathways. Thus, we have expanded the experiment for TGF- β -induced P4HA3 expression and demonstrated the function of P4HA3 on intrinsic EMT status or phenotypes in Fig. 4.

12. Authors mentioned, "Intriguingly, even though P4HA3 is the only enzyme involved in proline metabolism, its knockdown extensively abrogated TGF- β -altered amino acid metabolism." on Page 8, Lane 213-214, however, I think it was not provided as Figure (although it is mentioned in Supple Figure 3B). Could you show the change of amino acid level after knockdown or overexpression of P4HA3?

Response#3.12:

In Fig. 3G in the original manuscript (Fig. 3g in the revised manuscript), we show that *P4HA3* knockdown suppresses TGF- β induced amino acid changes. The figure number has been added to the above text in the revised manuscript (page 9, line 18).

13. Authors mentioned, "However, the function and mechanism of metabolites that promote EMT remains largely unknown." on Page 4, Lane 97-98, and this current study still did not show the molecular mechanism. The experiments with P4HA3 are quite interesting, but the relationship between the change of amino acids level by TGF- β and P4HA3 was not described here. (However, I agree with what authors said that "Further studies are necessary to reveal the molecular mechanisms of P4HA3 or proline metabolism that widely regulate amino acid metabolism, and also how TGF β -induced loss of amino acid promotes EMT status in cancer cells.")

Response#3.13:

We have removed the indicated text, and have improved the introduction section of the revised manuscript (page 4, lines 21–24).

14. Figure 1 and 2 describe the change of amino acid level by TGF- β and EMT, and Figure 3 and 4 focus on P4HA3 and EMT. The connection between these groups is somewhat weak, which should be more organised.

Response#3.14:

We appreciate this comment. As we believe that the amino acid reprogramming results during EMT, and its impact on EMT phenotypes, is particularly noteworthy, we arranged these results to Figs. 1

and 2. To further investigate the regulatory mechanisms associated with amino acid metabolism in EMT we identified the *P4HA3* gene in omics analysis and demonstrated its correlation with the EMT phenotypes *in vitro* and *in vivo* (Fig. 3 and 4). Although there are certainly other ways to organize the figures, we believe this version to be rationale and effective.

Minor points:

1. In Supple Figure 1G-I, PCA result of HCC827 seems to be opposite of A549 and H358, even Supple Figure 1A-F show similar results.

Response#3.15:

These PCA score plots (Supplementary Figs. 1g-i) were calculated based on different PCA loading factors in each dataset. Thus, the location of plots in the PCA score show different metabolic features in each figure. We have rechecked the PCA and confirmed that they are accurate.

2. In Figure 2B, legend and labelling are a little bit confusing. What does this “Unstimulated” mean in the right panel? I thought the withdrawal was performed after TGF- β treatment. As described in the legend, if this unstimulated is just normal growth media, then how did authors compare with the result after the withdrawal? a. Withdraw: I believe this was done like TGF- β induction for 3 days, and changed with normal media, then incubation for further 3 days. b. Unstimulated: So, this is just normal media for 6 days?

Response#3.16:

We thank the reviewer for this helpful comment and have revised the figure legend as follows (page 24, lines 5–8):

“Controls were cultured in normal growth medium for 3 days (‘Unstimulated (A)’) and 6 days (‘Unstimulated (B)’) for TGF- β stimulation and withdrawal, respectively. The medium for ‘Unstimulated (B)’ was changed to fresh normal medium after culturing for 3 days, as well as for ‘withdraw’ cultures.”

3. In Figure 4F, it is a little bit difficult to see the metastatic foci in the left image. (Not sure whether it is due to the low resolution of the image)

Response#3.17:

We agree with the reviewer’s comment. Considering the low resolution of this image we have

removed it from the revised manuscript.

4. *On Page 8, Lane 216, authors mentioned “Figure 6”. Where are Figure 5 and 6?*

Response#3.18:

I have corrected the figure number, “Figure 6” to “Fig. 2” (Page 9, line 20).

Reviewers' Comments:

Reviewer #1:

Remarks to the Author:

In the revised version of their manuscript, the authors have now addressed all my initial concerns. The manuscript has been greatly improved and provides a very consistent set of data.

Reviewer #2:

Remarks to the Author:

Thank you for taking the time to carefully consider suggestions and comments. The revised manuscript is strengthened by the new data included, and the discussions added have expanded the scope of findings presented.

Reviewer #3:

Remarks to the Author:

The authors have done excellent work and answered everything I had addressed. I am quite satisfied and I support the publication of this revised manuscript.